# Boosted charge and proton transfer over ternary Co/Co$_3$O$_4$/CoB for electrochemical nitric oxide reduction to ammonia

Xiaoxuan Fan[1], Zhenyuan Teng[2], Lupeng Han [1] ✉, Yongjie Shen[3], Xiyang Wang [4], Wenqiang Qu[1], Jialing Song[1], Zhenlin Wang[1], Haiyan Duan[1], Yimin A. Wu [4], Bin Liu [2,5] ✉ & Dengsong Zhang [1] ✉

The electrochemical nitric oxide reduction reaction (NORR) holds a great potential for removing environmental pollutant NO and meanwhile generating high value-added ammonia (NH$_3$). Herein, we tactfully design and synthesize a ternary Co/Co$_3$O$_4$/CoB heterostructure that displays a high NH$_3$ Faradaic efficiency of 98.8% in NORR with an NH$_3$ yield rate of 462.18 μmol cm$^{-2}$ h$^{-1}$ (2.31 mol h$^{-1}$ g$_{cat}^{-1}$) at −0.5 V versus reversible hydrogen electrode, outperforming most of the reported NORR electrocatalysts to date. The superior NORR performance is attributed to the enhanced charge and proton transfer over the ternary Co/Co$_3$O$_4$/CoB heterostructure. The charge transfer between CoB and Co/Co$_3$O$_4$ yields electron-deficient Co and electron-rich Co$_3$O$_4$. The electron-deficient Co sites boost H$_2$O dissociation to generate *H while the electron-rich low-coordination Co$_3$O$_4$ sites promote NO adsorption. The *H formed on electron-deficient Co sites is more favorable to transfer to electron-rich Co$_3$O$_4$ sites adsorbed with NO, facilitating the selective hydrogenation of NO. This study paves the way for designing and developing highly efficient electrocatalysts for electrochemical reduction of NO to NH$_3$.

Ammonia (NH$_3$) is an indispensable chemical for agriculture, chemical industry, refrigerant, hydrogen carrier, and healthcare[1–5]. Unfortunately, NH$_3$ production by traditional Haber-Bosch method requires harsh conditions of high temperature and high pressure, emitting a large quantity of greenhouse gases[6,7]. One of the alternatives to the Haber-Bosch method for NH$_3$ synthesis is the electrocatalytic reduction of nitrogen-containing species, including nitrogen gas (N$_2$), nitric oxide (NO), nitrite (NO$_2^-$), and nitrate (NO$_3^-$)[8–11]. The nitrogen reduction reaction (NRR) suffers from low NH$_3$ Faradaic efficiency (FE$_{NH_3}$) and NH$_3$ yield rate because of the high dissociation energy (941 kJ mol$^{-1}$) of N≡N and the competitive hydrogen evolution

reaction (HER) (the reduction potential of N$_2$ (0.093 V vs. RHE) and H$_2$O (0 V vs. RHE) are close to each other)[12–14]. The electroreduction of NO$_2^-$ and NO$_3^-$ to NH$_3$ (NO$_2^-$RR and NO$_3^-$RR) have more complicated reaction pathways, and generate more types of N-containing byproducts[15,16]. Compared with NRR, the electrochemical NO reduction reaction (NORR) is kinetically and thermodynamically more favorable because NO possesses a lower dissociation energy (204 kJ mol$^{-1}$) and a more positive reduction potential (0.71 V vs. RHE)[17–19]. The approach to couple NO oxidation with NO$_3^-$ reduction is also considered as an alternative to NORR. But NORR to produce NH$_3$ generally requires lower energy consumption[20,21]. In recent years, NORR is becoming

[1]Innovation Institute of Carbon Neutrality, International Joint Laboratory of Catalytic Chemistry, State Key Laboratory of Advanced Special Steel, Department of Chemistry, College of Sciences, Shanghai University, Shanghai, China. [2]Department of Materials Science and Engineering, City University of Hong Kong, Hong Kong SAR, China. [3]Institute for Chemical Reaction Design and Discovery (WPI-ICReDD), Hokkaido University, Sapporo, Japan. [4]Department of Mechanical and Mechatronics Engineering, Waterloo Institute for Nanotechnology, University of Waterloo, ON Waterloo, Canada. [5]Department of Chemistry, Hong Kong Institute of Clean Energy (HKICE) & Center of Super-Diamond and Advanced Films (COSDAF), City University of Hong Kong, Hong Kong SAR, China. ✉e-mail: lphan@shu.edu.cn; bliu48@cityu.edu.hk; dszhang@shu.edu.cn

increasingly attractive because NORR not only can remove environmental pollutant NO from industrial exhaust gas, but also produces value-added chemicals.

The electrochemical reduction of NO involves chemical activation of NO followed by NO hydrogenation[22–24]. Currently, the $FE_{NH3}$ and $NH_3$ yield rate in electrochemical NORR are still impeded by the weak adsorption of NO and the sluggish proton supply. To improve the NORR performance, researchers have made endeavors to enhance the adsorption of NO by transition metal oxide modification or vacancy engineering[25,26]. The proton supply could be enhanced by using metal-based electrocatalysts that would facilitate $H_2O$ dissociation[24,27]. Constructing single atoms in amorphous metal oxides with oxygen vacancies to form metal-O moieties could accelerate hydrogenation of NO to produce $NH_3$[28–30]. However, till now, it is still intractable to acquire both high $FE_{NH3}$ and $NH_3$ yield rate simultaneously because of the complex balance of the NO adsorption and $H_2O$ dissociation. Early studies have shown fairly high NO adsorption capacity over $Co_3O_4$ surface[31–34]. However, the poor activation and dissociation of $H_2O$ on $Co_3O_4$ greatly restricts the hydrogenation reaction in NORR[35]. Metallic Co on $Co_3O_4$ heterostructure is designed to promote $H_2O$ dissociation to increase available *H[36]. But this concurrently accelerates HER, leading to a low NORR $FE_{NH3}$ and $NH_3$ yield rate[27,30,37–39]. Tuning charge distribution on catalyst's surface is able to directionally drive *H transfer to enhance NO hydrogenation during NORR[40,41]. Boron (B) possesses a $2s^2p^1$ electronic structure, which exhibits flexible valence states during catalytic reactions. Introducing B into the Co/$Co_3O_4$ heterostructure is anticipated to optimize NO adsorption and $H_2O$ dissociation, and in the meantime induce *H transfer to promote NO hydrogenation and suppress HER during NORR, thus greatly enhancing NORR activity and selectivity.

In this study, we tactfully designed and synthesized a ternary Co/$Co_3O_4$/CoB heterostructure by reducing $Co_3O_4$ with $NaBH_4$. The ternary Co/$Co_3O_4$/CoB catalyst exhibited an excellent electrochemical

NORR performance, achieving an $NH_3$ yield rate of 462.18 $\mu mol\ cm^{-2}\ h^{-1}$ (2.31 $mol\ h^{-1}\ g_{cat}^{-1}$) and an $FE_{NH3}$ of 98.80% at −0.5 V vs. RHE, outperforming most of the reported NORR electrocatalysts to date. To demonstrate the application potential, a Zn-NO battery was assembled using the Co/$Co_3O_4$/CoB as the cathode, which delivered a high power density of 10.06 $mW\ cm^{-2}$. A series of experiments and density functional theory (DFT) calculations revealed that the enhanced charge and proton transfer over the ternary Co/$Co_3O_4$/CoB effectively boosted the electrochemical NORR. In detail, the charge transfer between CoB and Co/$Co_3O_4$ yielded electron-deficient Co and electron-rich $Co_3O_4$. The electron-deficient Co sites boosted $H_2O$ dissociation to generate *H, while the electron-rich low-coordination $Co_3O_4$ sites promoted NO adsorption and *H transfer. Thanks to the enhanced charge and proton transfer, the energy barrier of the NORR potential-determining step from *NO to *HNO over the ternary Co/$Co_3O_4$/CoB was greatly reduced. Furthermore, the Co/$Co_3O_4$/CoB could also facilitate adsorption of *NH and *$NH_2$, beneficial for NO reduction to produce $NH_3$.

## Results
### Catalyst synthesis and structure characterization
The ternary Co/$Co_3O_4$/CoB heterostructured electrocatalyst was synthesized by reducing $Co_3O_4$ using $NaBH_4$ in an Ar atmosphere at 500 °C for 4 h (Fig. 1a). By applying the same method, Co/$Co_3O_4$ could also be obtained by decreasing the quality ratio of $NaBH_4$/$Co_3O_4$. The as-prepared Co/$Co_3O_4$/CoB shows the morphology of nanoparticles supported on nanosheets based on the transmission electron microscope (TEM, Supplementary Fig. 1) measurement. The average thickness of the nanosheets is ~3.3 nm determined by atomic force microscopy (AFM, Fig. 1b). The nanoparticles on the nanosheets are composed of large nanoparticles surrounded by many small nanoparticles (HRTEM, Supplementary Fig. 2). Co/$Co_3O_4$/CoB shows $Co_3O_4$

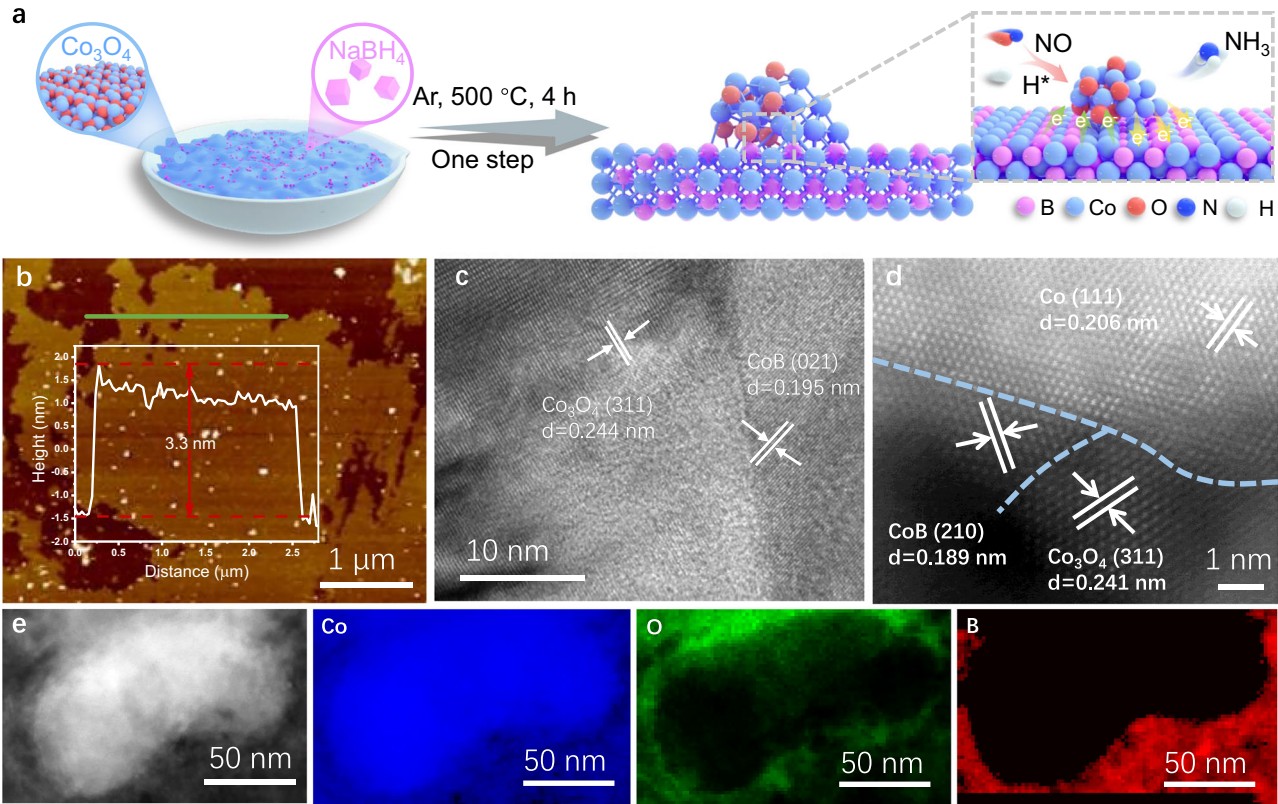

**Fig. 1 | Synthesis and morphology of Co/$Co_3O_4$/CoB. a** Schematic diagram showing the synthesis process of Co/$Co_3O_4$/CoB. **b** AFM image of Co/$Co_3O_4$/CoB. The inset shows the height profile along the red line in (**b**). **c** HRTEM image of Co/

$Co_3O_4$/CoB. **d** Aberration-corrected HAADF-STEM image of Co/$Co_3O_4$/CoB. **e** HAADF-STEM image and the corresponding EELS mappings of Co, O, and B elements in Co/$Co_3O_4$/CoB. Source data for Fig. 1 are provided as a Source Data file.

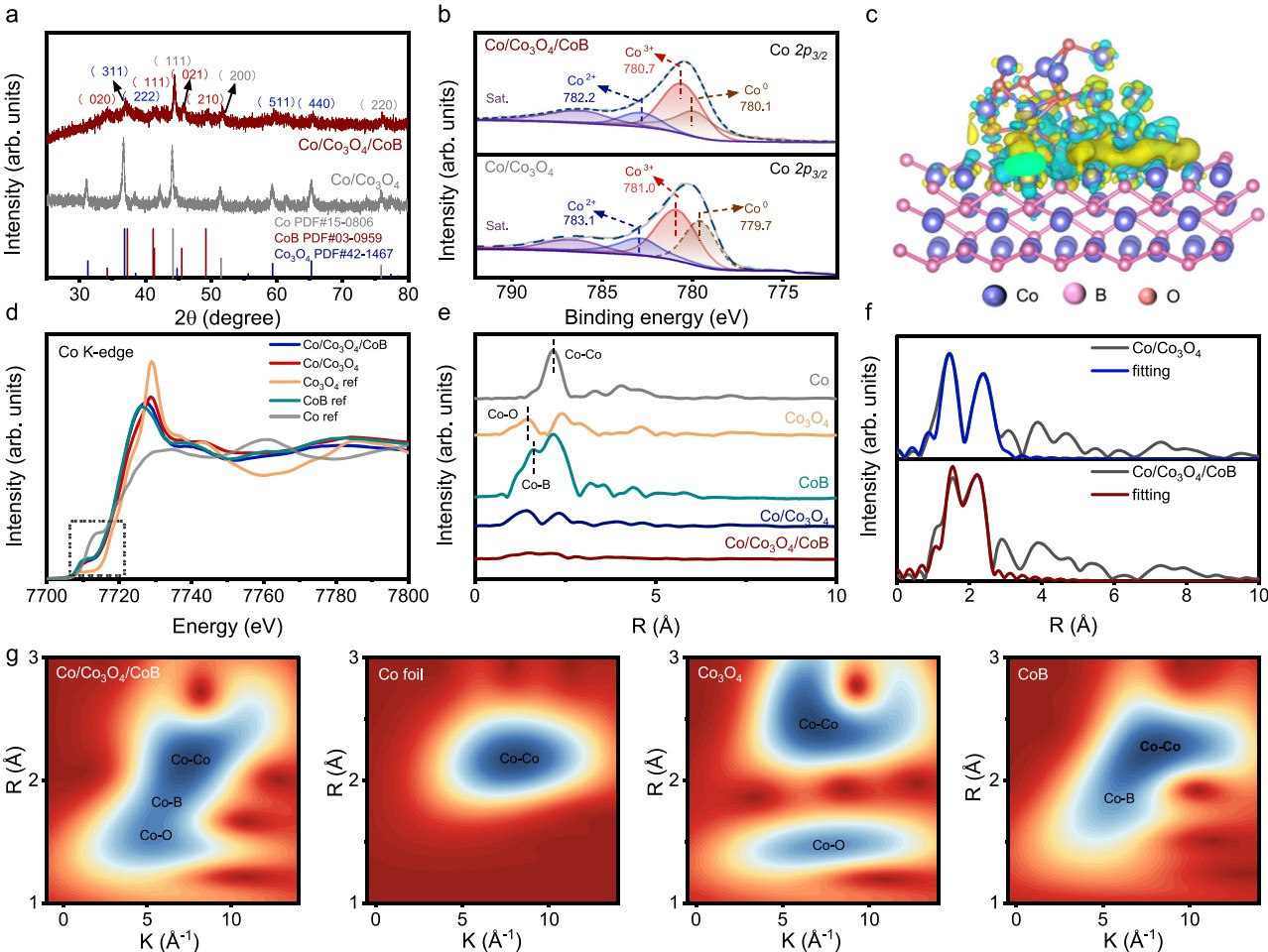

**Fig. 2 | Structural features of Co/Co₃O₄/CoB. a** XRD patterns of Co/Co₃O₄ and Co/Co₃O₄/CoB. **b** Co $2p_{3/2}$ XPS spectra of Co/Co₃O₄ and Co/Co₃O₄/CoB. **c** The charge density difference analysis for Co/Co₃O₄/CoB. Yellow and blue colors represent electron accumulation and depletion regions, respectively. The isosurface value is 0.004 e/Å³. **d** Co K-edge XANES spectra of Co/Co₃O₄, Co/Co₃O₄/CoB, Co foil, CoB, and Co₃O₄. **e** The FT-EXAFS spectra of Co/Co₃O₄, Co/Co₃O₄/CoB, Co foil, CoB, and Co₃O₄. **f** FT-EXAFS fitting curves for Co/Co₃O₄ and Co/Co₃O₄/CoB in R space. **g** WT contour plots of Co/Co₃O₄/CoB, Co foil, CoB, and Co₃O₄. Source data for Fig. 2 are provided as a Source Data file.

(311) and CoB (021) crystalline phases from the HRTEM image (Fig. 1c). To precisely determine the structure of Co/Co₃O₄/CoB, aberration-corrected HAADF-STEM measurement was further performed. As shown in Fig. 1d, abundant interfaces among Co, Co₃O₄, and CoB phases can be clearly observed, suggesting the formation of a ternary Co/Co₃O₄/CoB heterostructure. The electron energy loss spectroscopy (EELS) mapping (Fig. 1e) combined with the HRTEM image (Supplementary Fig. 2) clarifies that the Co/Co₃O₄/CoB heterostructure is composed of a CoB nanosheet supported with large Co and small Co₃O₄ nanoparticles. For comparison, Co/Co₃O₄ shows a heterostructure of Co nanoparticles supported on Co₃O₄ nanosheets (Supplementary Figs. 3 and 4).

The structure and valence state of Co/Co₃O₄/CoB were analyzed by X-ray powder diffraction (XRD) and X-ray photoelectron spectroscopy (XPS). As shown in Fig. 2a, the XRD patterns show clear diffraction peaks of Co₃O₄ (PDF #42-1467), Co (PDF #15-0806) and CoB (PDF #03-0959) for Co/Co₃O₄/CoB, and Co₃O₄ (PDF #42-1467) and Co (PDF #15-0806) for Co/Co₃O₄. The B *1s* XPS spectrum of Co/Co₃O₄/CoB also evidences the formation of CoB (Supplementary Fig. 5). In Fig. 2b, the Co $2p_{3/2}$ XPS spectrum of Co/Co₃O₄ exhibits three peaks at binding energies 779.7, 781.0 and 783.1 eV, which can be attributed to Co⁰, Co³⁺ and Co²⁺, respectively[42–44]. For Co/Co₃O₄/CoB, the Co⁰ peak slightly shifts to a higher binding energy (780.1 eV), while the Co³⁺ (780.7 eV) and Co²⁺ (782.2 eV) peaks shift toward lower binding

energies as compared to those of Co/Co₃O₄. Because Co⁰ and Co³⁺/Co²⁺ mainly exist in Co and Co₃O₄, respectively, it can be inferred that the introduction of CoB leads to more electron loss in Co and more electron gain in Co₃O₄. To clarify the electron transfer among different interfaces in Co/Co₃O₄/CoB and Co/Co₃O₄, DFT calculations were conducted. Herein, by balancing the computation power and calculation accuracy, the Co/Co₃O₄/CoB was modeled by Co (111) and Co₃O₄ (311) clusters supported on CoB (021), while Co/Co₃O₄ was modeled by Co (111) clusters supported on Co₃O₄ (311) (for details see DFT calculations in Method section). Firstly, the Bader charge was analyzed over Co/Co₃O₄ and Co/Co₃O₄/CoB models. For the Co/Co₃O₄ model, 3.64 |e| is transferred from Co to Co₃O₄ (Supplementary Fig. 6, Supplementary Table 1, and Supplementary Data 1). In Co/Co₃O₄/CoB, Co₃O₄ gets 2.23 |e| and 2.36 |e| from Co and CoB, respectively (Supplementary Fig. 7, Supplementary Table 1, and Supplementary Data 1). The charge density difference analysis of Co/Co₃O₄/CoB indicates a significant charge redistribution at the interface between CoB and Co/Co₃O₄ (Fig. 2c), with electron transfer from Co to CoB and from CoB to Co₃O₄. Furthermore, the average numbers of electron transfer per Co atom in Co/Co₃O₄/CoB and Co/Co₃O₄ were analyzed. As compared in Supplementary Table 1, Co loses more electrons (−0.22 e vs. −0.13 e) and Co₃O₄ gains more electrons (+0.51 e vs. +0.08 e) in Co/Co₃O₄/CoB as compared to Co/Co₃O₄, evidencing that CoB introduction can modulate the electronic structure of Co and Co₃O₄ in Co/Co₃O₄/CoB. X-ray

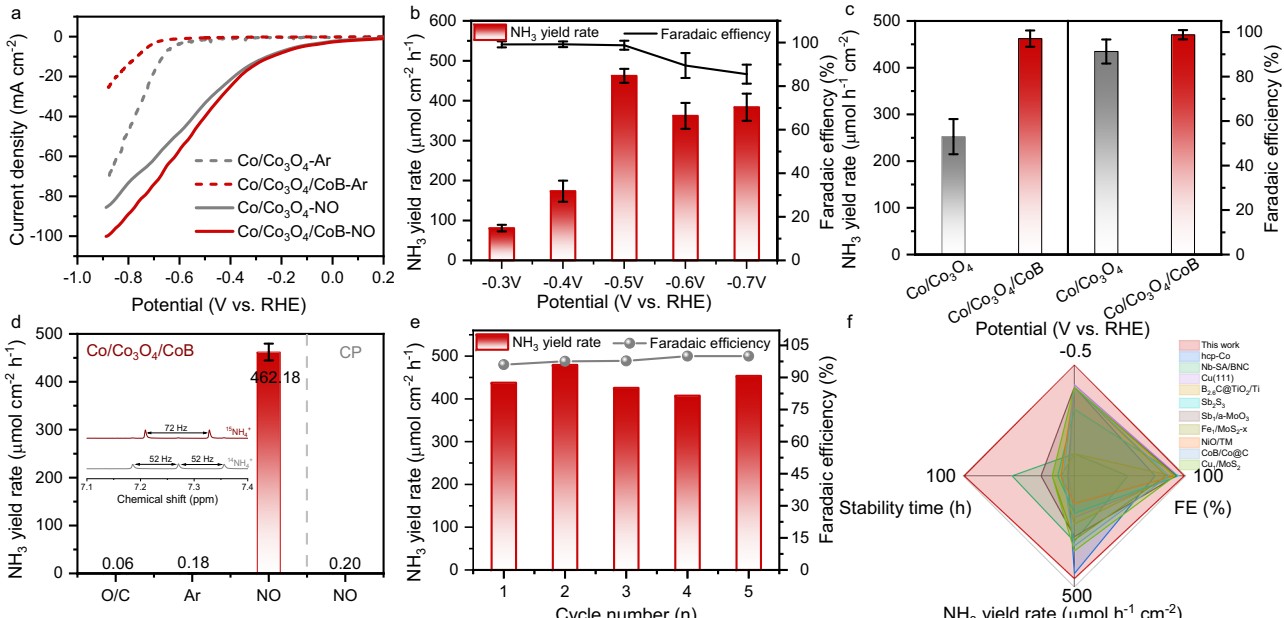

**Fig. 3 | Electrochemical NORR performance. a** LSV curves of Co/Co$_3$O$_4$ and Co/Co$_3$O$_4$/CoB with a scanning of 5 mV/s recorded in 0.1 M Ar-saturated and NO-saturated PBS (pH = 7.3 ± 0.1, with 90% iR-correction). **b** FE$_{NH3}$ and NH$_3$ yield rate over Co/Co$_3$O$_4$/CoB at various cathodic potentials with a mass loading of 0.2 mg cm$^{-2}$ at 25 °C. **c** Comparison of NORR performance between Co/Co$_3$O$_4$ and Co/Co$_3$O$_4$/CoB at -0.5 V vs. RHE. **d** NORR performance over Co/Co$_3$O$_4$/CoB and carbon paper (CP) under different conditions (O/C represents open circuit

condition). The inset shows the $^1$H nuclear magnetic resonance spectra to quantify NH$_4^+$. **e** Stability test of Co/Co$_3$O$_4$/CoB at −0.5 V vs. RHE. **f** Comparison of electro-catalytic NORR performance of Co/Co$_3$O$_4$/CoB with other reported electrocatalysts in literature. The error bars denote the standard deviation derived from three independent measurements, with the central marker indicating the mean value. Source data for Fig. 3 are provided as a Source Data file.

absorption spectroscopy (XAS), including X-ray absorption near-edge structure (XANES) and extended X-ray absorption fine structure (EXAFS), was performed to examine the electronic structure and coordination environment of Co in Co/Co$_3$O$_4$/CoB and Co/Co$_3$O$_4$. In the Co K-edge XANES spectra (Fig. 2d), the position of the absorption edge for Co/Co$_3$O$_4$/CoB and Co/Co$_3$O$_4$ are located between that for Co foil and Co$_3$O$_4$ references, indicating that the average valence state of Co atom in Co/Co$_3$O$_4$/CoB and Co/Co$_3$O$_4$ are within 0 and +2/+3, owing to co-existence of both metallic Co and Co$_3$O$_4$. To understand the local structure around Co atom in these samples, the EXAFS spectra are transformed into R-space. The R-space EXAFS spectra (Fig. 2e) display dominant peaks of Co-B (1.60 Å)[45], Co-Co (2.17 Å)[46], and Co-O (1.44 Å)[47] coordination bonds in CoB, Co foil and Co$_3$O$_4$, respectively. The FT-EXAFS spectra of Co/Co$_3$O$_4$/CoB and Co/Co$_3$O$_4$ can be well-fitted using backscattering paths of Co-Co, Co-B and Co-O (Fig. 2f, Supplementary Tables 2 and 3). Notably, the Co-O coordination number (2.2) in Co/Co$_3$O$_4$/CoB is significantly lower than that (5.7) in Co/Co$_3$O$_4$. The low-coordinated Co-O structure in Co/Co$_3$O$_4$/CoB is beneficial for NO adsorption[48]. The wavelet transform (WT) contour plot of Co/Co$_3$O$_4$/CoB (Fig. 2g) displays three intensity maximums at around 7.2, 5.9, and 5.4 Å$^{-1}$, corresponding to the Co-Co, Co-B, and Co-O coordination, respectively[47,49]. For comparison, Co/Co$_3$O$_4$ only shows Co-Co and Co-O coordinations (Supplementary Fig. 8).

## Electrochemical NORR performance

The electrochemical NO reduction performance over Co/Co$_3$O$_4$/CoB was assessed in an H-type electrolytic cell filled with 0.1 M phosphate buffer saline (PBS) aqueous solution. As shown in Fig. 3a and Supplementary Fig. 9, the linear sweep voltammetry (LSV) curves (iR-corrected or not) reveal that both Co/Co$_3$O$_4$/CoB and Co/Co$_3$O$_4$ exhibit a significantly increased current density in 10 vol.% NO/Ar atmosphere as compared to pure Ar atmosphere, suggesting preferred NORR on Co/Co$_3$O$_4$/CoB and Co/Co$_3$O$_4$ surface. Chron-oamperometric (1 h electrolysis) and ultraviolet-visible (UV-vis)

colorimetric measurements were used to quantitatively determine the NH$_3$ Faradaic efficiency (FE) and yield rate (Supplementary Figs. 10 and 11). At −0.5 V vs. RHE, the Co/Co$_3$O$_4$/CoB displays the optimum FE$_{NH3}$ of 98.80% and NH$_3$ yield rate of 462.18 μmol h$^{-1}$ cm$^{-2}$ (2.31 mol h$^{-1}$ g$_{cat}^{-1}$), much higher than Co/Co$_3$O$_4$ (FE$_{NH3}$ of 91.28% and NH$_3$ yield rate of 252.34 μmol h$^{-1}$ cm$^{-2}$, Fig. 3b, c). $^1$H nuclear magnetic resonance (NMR) measurements give comparable results (Supplementary Figs. 12–14). The possible by-products produced over Co/Co$_3$O$_4$/CoB during NORR were checked by differential electrochemical mass spectroscopy (DEMS), UV-vis absorption spectroscopy, and gas chromatography (Supplementary Figs. 15–18). The hydroxylamine (NH$_2$OH), N$_2$ and N$_2$O by-products were not detected. Moreover, we conducted the NORR experiment over Co/Co$_3$O$_4$/CoB in 0.1 M KOH aqueous electrolyte. The NORR performance was poorer as compared with the experiment conducted in 0.1 M PBS (Supplementary Fig. 19). The better NORR performance of Co/Co$_3$O$_4$/CoB as compared to that of Co/Co$_3$O$_4$ can be further reflected by the smaller semicircle radius in the electrochemical impedance spectroscopy (EIS) spectrum (Supplementary Fig. 20), indicating a smaller charge transfer resistance (R$_{ct}$) over Co/Co$_3$O$_4$/CoB surface. A series of control experiments indicate that the nitrogen in the NH$_3$ product originates solely from NO (Fig. 3d). Carbon paper itself shows negligible NORR activity, disclosing that NH$_3$ is mainly produced on Co/Co$_3$O$_4$/CoB (Fig. 3d). Using isotopic $^{15}$NO as the reactant, the characteristic double peaks (inset in Fig. 3d) appear with a separation of 72 Hz, corresponding to $^{15}$NH$_4^+$, indicating that $^{15}$NO is the sole nitrogen source of NH$_3$ production. Besides excellent catalytic activity, Co/Co$_3$O$_4$/CoB also displays good catalytic stability. During five consecutive electrolysis cycles at −0.5 V vs. RHE, the FE$_{NH3}$ and NH$_3$ yield rate remain nearly unchanged (Fig. 3e). Moreover, this catalyst exhibits good current stability for ~100 h at −0.5 V vs. RHE (Supplementary Fig. 21). Co/Co$_3$O$_4$/CoB shows a good structural stability during the NORR based on the XRD, HRTEM and XPS characterizations (Supplementary Figs. 22–24). The ternary Co/Co$_3$O$_4$/CoB achieves a high FE$_{NH3}$ of 98.80% and NH$_3$ yield rate of

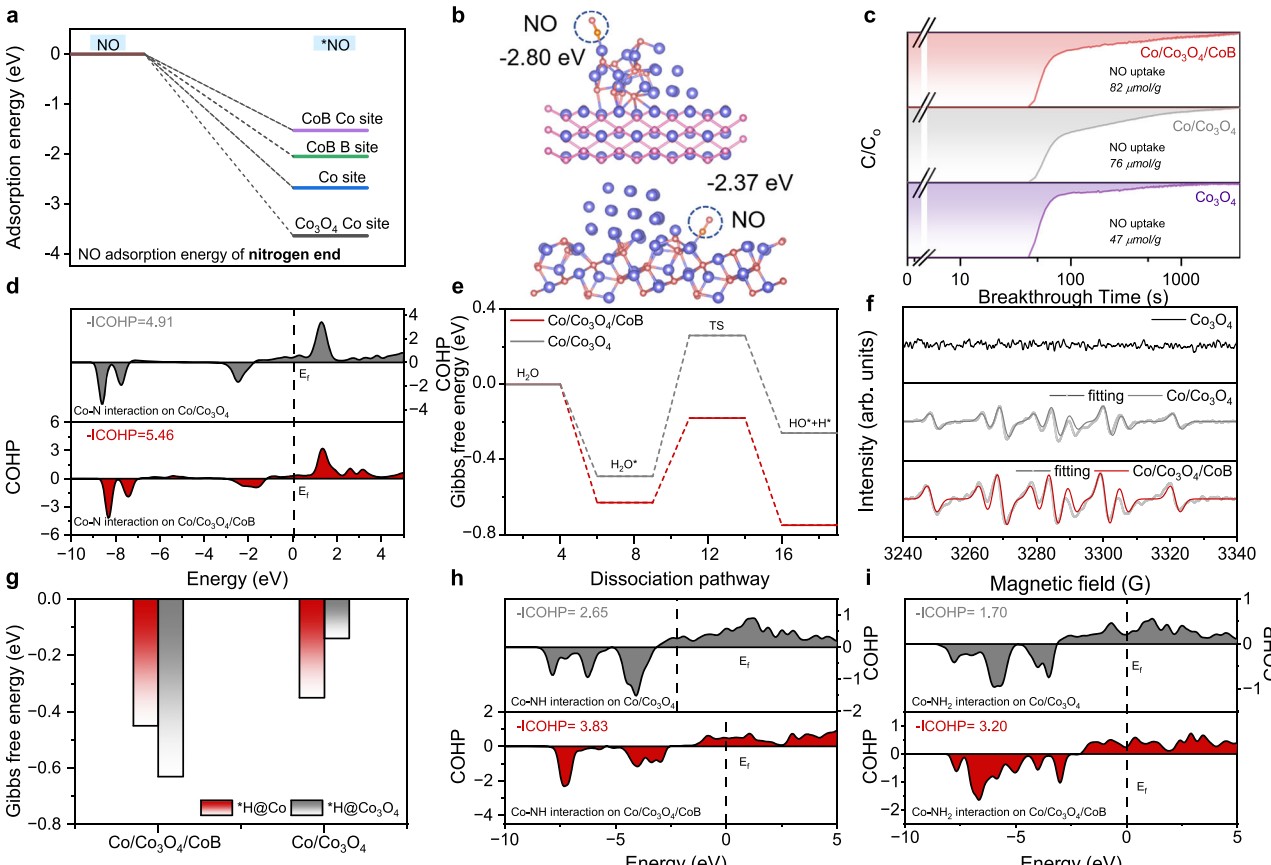

**Fig. 4 | Reaction mechanism. a** The calculated NO adsorption energy of nitrogen end on Co (111), CoB (021) and $Co_3O_4$ (311). **b** The calculated NO adsorption energy on $Co/Co_3O_4/CoB$ and $Co/Co_3O_4$. The blue circle marks the NO molecule. Atom color-coding: Purple, cobalt; red, oxygen; pink, boron; nitrogen, orange. **c** NO adsorption breakthrough curves of $Co/Co_3O_4/CoB$, $Co/Co_3O_4$ and $Co_3O_4$ catalysts at 25 °C. **d** The COHP analysis of Co-N bond during NO adsorption on $Co/Co_3O_4/CoB$ and $Co/Co_3O_4$. **e** Gibbs free energy for $H_2O$ dissociation on $Co/Co_3O_4/CoB$ and $Co/Co_3O_4$. **f** EPR spectra recorded over $Co/Co_3O_4/CoB$, $Co/Co_3O_4$ and $Co_3O_4$ upon electrolysis in the absence of NO. **g** Comparison of Gibbs free energy for *H adsorption on Co and $Co_3O_4$ sites of $Co/Co_3O_4/CoB$ and $Co/Co_3O_4$. The COHP analysis of **h** *NH and **i** $*NH_2$ on $Co/Co_3O_4/CoB$ and $Co/Co_3O_4$. Source data for Fig. 4 are provided as a Source Data file.

462.18 µmol h⁻¹ cm⁻² as well as long-time stability at a low cathodic potential of −0.5 V vs. RHE, superior to most of the reported NORR electrocatalysts (Fig. 3f and Supplementary Tables 4 and 5). To demonstrate the practical application potential of the $Co/Co_3O_4/CoB$ catalyst, a Zn-NO battery with $Co/Co_3O_4/CoB$ cathode and Zn plate anode was assembled (Supplementary Fig. 25), which could deliver an open circuit voltage (OCV) of 2.04 V (Supplementary Fig. 26) and a maximum power density of 10.06 mW cm⁻², outperforming all reported results in the literature (Supplementary Figs. 27 and 28, Supplementary Table 6). The output discharge current density increases continuously from 0.5 to 8 mA cm⁻² and each step exhibits a stable discharging plateau, indicating that our cell has an excellent discharge capability (Supplementary Fig. 29). Moreover, the $NH_3$ yield rate exhibits a maximum of 1627.67 µg h⁻¹ mg$_{cat}$⁻¹ at 8 mA cm⁻² (Supplementary Fig. 30). Significantly, the Zn-NO battery can remove NO pollutant, produce $NH_3$ and generate electricity at the same time.

## Reaction mechanism

To understand the excellent NORR performance of $Co/Co_3O_4/CoB$, Brunauer-Emmett-Teller (BET) surface area and electrochemically active surface area (ECSA) of the as-prepared catalysts were first measured. Compared to $Co/Co_3O_4$, $Co/Co_3O_4/CoB$ displays a higher BET surface area (Supplementary Fig. 31) and ECSA (Supplementary Fig. 32), which shall offer more active sites to catalyze electrochemical NORR. To clarify the reactive sites for NO adsorption/activation and $H_2O$ dissociation as well as the protonation process of adsorbed intermediates

on $Co/Co_3O_4/CoB$, a series of DFT calculations, NO breakthrough and electron paramagnetic resonance (EPR) measurements were conducted. The NO adsorption energies with N-end/O-end on the Co sites of Co (111), $Co_3O_4$ (311), and CoB (021) and the B sites of CoB (021) were computed by DFT (Fig. 4a and Supplementary Figs. 33–35). The results indicate that NO prefers to adsorb on Co sites of $Co_3O_4$ (311) in the form of N-end adsorption, having the most negative adsorption energy of −3.64 eV. Compared to $Co/Co_3O_4$, $Co/Co_3O_4/CoB$ binds to NO more strongly (Fig. 4b and Supplementary Data 1). The low-coordinated Co-O structure in $Co/Co_3O_4/CoB$ (Fig. 2f and Supplementary Table 2) contributes to NO adsorption as demonstrated by NO breakthrough analysis (Fig. 4c)[47]. A crystal orbital Hamilton population (COHP) was further employed to investigate the NO adsorption strength on $Co/Co_3O_4/CoB$ and $Co/Co_3O_4$ (Fig. 4d)[50–52]. Compared to $Co/Co_3O_4$, $Co/Co_3O_4/CoB$ displays a more negative integrated COHP (ICOHP) value of the Co-N bond, indicating a stronger NO adsorption on the $Co/Co_3O_4/CoB$ surface[53]. DFT calculation was further performed to explore the water dissociation process. The energy barrier for water transformation from the adsorbed state ($H_2O^*$) to the transition state (TS) on $Co/Co_3O_4/CoB$ and $Co/Co_3O_4$ are 0.50 eV and 0.66 eV, respectively (Fig. 4e, Supplementary Figs. 36 and 37, Supplementary Table 6). This result indicates that the dissociation of $H_2O$ is more kinetically favorable on $Co/Co_3O_4/CoB$. In order to further investigate the $H_2O$ dissociation ability of $Co/Co_3O_4/CoB$ and $Co/Co_3O_4$, in-situ Raman spectroscopy measurements were conducted at different applied cathodic potentials. Three types of O-H stretching modes belonging to

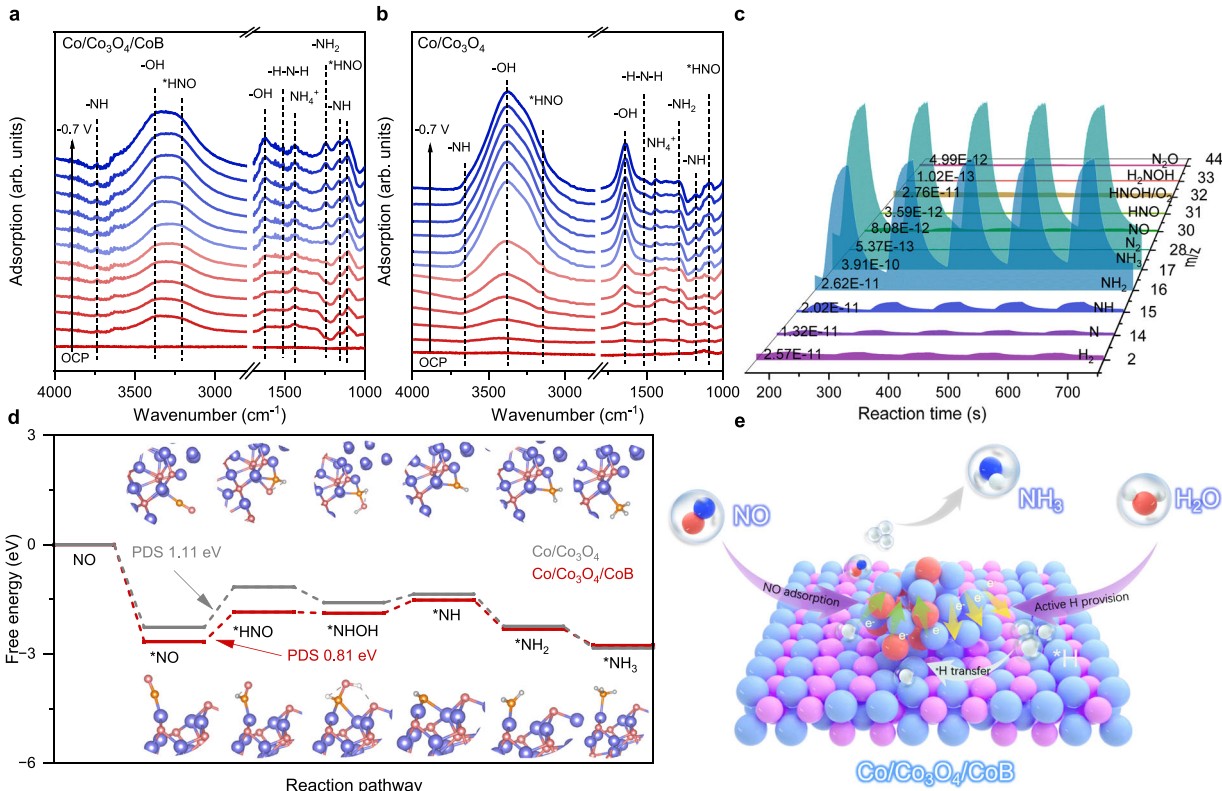

**Fig. 5 | Reaction pathway.** In-situ ATR-SEIRAS spectra recorded over Co/Co₃O₄/CoB **a** and Co/Co₃O₄ **b** in 0.1 M PBS at the applied cathodic potential from OCP to −0.7 V vs. RHE. **c** Online DEMS measurement results over Co/Co₃O₄/CoB at −0.5 V vs. RHE. **d** The calculated NORR Gibbs free energy diagrams over Co/Co₃O₄ and Co/Co₃O₄/ CoB at 0 V vs. RHE. Atom color-coding: Purple, cobalt; red, oxygen; orange, nitrogen;

gray, hydrogen. **e** Schematic illustration showing the NORR mechanism over Co/ Co₃O₄/CoB electrocatalyst. Atom color-coding: Light blue, cobalt; pink, boron; red, oxygen; deep blue, nitrogen; gray, hydrogen. Source data for Fig. 5 are provided as a Source Data file.

4-HB·H₂O, 2-HB·H₂O, and M·H₂O (M = Na⁺/K⁺), respectively[54], are observed. The relative proportion of M·H₂O displays a little increasing trend along with increasing the applied cathodic potential, suggesting that M·H₂O is closer to the catalyst's surface and could be more affected by the electric field. The M·H₂O proportion over Co/Co₃O₄/ CoB is always higher than that over Co/Co₃O₄ at all studied cathodic potentials, suggesting better water dissociation ability over Co/Co₃O₄/ CoB (Supplementary Figs. 38 and 39)[55,56]. Consequently, Co/Co₃O₄/CoB could generate more *H as reflected in the liquid EPR measurement using 5,5-dimethyl-1-pyrroline-N-oxude (DMPO) as the *H trapping regent (Fig. 4f)[57]. The DMPO-H signal completely disappeared when NO was introduced into the reaction system (Supplementary Fig. 40), indicating that the generated *H could be rapidly consumed by NO on the Co/Co₃O₄/CoB surface. This further verifies that *H is more favorable for bonding with NO to form NHₓ intermediates rather than generating H₂[58]. The *H generated on the metallic Co sites in Co/Co₃O₄/CoB can be quickly transferred to the neighboring Co₃O₄ surface to trigger the *NO hydrogenation reaction (Fig. 4g), thanks to the relatively electron-richer Co₃O₄ in Co/Co₃O₄/CoB than that in Co/Co₃O₄. As a result, the enhanced charge and *H transfer over Co/Co₃O₄/CoB promote electrochemical NO reduction to produce NH₃. The COHP analysis was performed to investigate the adsorption capacity of nitrogen-containing intermediates on Co/Co₃O₄/CoB and Co/Co₃O₄. The ICOHP value of the Co-NH/Co-NH₂ bond is −3.83/3.20 and −2.65/1.70 on Co/ Co₃O₄/CoB and Co/Co₃O₄, respectively (Fig. 4h, i, Supplementary Figs. 41 and 42), indicating a higher bonding state energy of *NH/*NH₂ on Co/Co₃O₄/CoB than Co/Co₃O₄. The projected density of states (PDOS) of Co/Co₃O₄/CoB and Co/Co₃O₄ upon adsorption of *NH and *NH₂ were analyzed (Supplementary Figs. 43 and 44). The $E_p$ (the highest peak below the Fermi level) position of Co/Co₃O₄/CoB is higher

than that of Co/Co₃O₄ upon adsorption of *NH and *NH₂, indicating that the antibonding states of NORR intermediates are located at higher energies with lower occupancies on Co/Co₃O₄/CoB surface, which results in stronger bindings of intermediates on Co/Co₃O₄/CoB during NORR[59]. Both the COHP and PDOS results indicate enhanced adsorption of *NH and *NH₂ intermediates over Co/Co₃O₄/CoB, which is conducive to NORR for producing NH₃.

To reveal the reaction pathway of electrochemical NORR, in-situ attenuated total reflection surface enhanced infrared absorption spectroscopy (ATR-SEIRAS) measurements were conducted to probe the reaction intermediates in the potential range from OCP to −0.7 V (vs. RHE). Over the Co/Co₃O₄/CoB surface (Fig. 5a), the bands at 1103 and 3224 cm⁻¹ belong to *HNO species while the bands at 1167, 1245, and 1504 cm⁻¹ are related to N-H stretching, -NH₂ wagging, and -H-N-H bending, respectively[60,61]. The gradually increased bands at 1456 and 1635 cm⁻¹ can be attributed to NH₄⁺ and -OH species accumulation on the Co/Co₃O₄/CoB surface[62]. Additionally, a wide overlapped absorption at around 3000 - 3800 cm⁻¹ is observed, in which the bands at 3380 and 3731 cm⁻¹ are assigned to the stretching of -OH and -NH, respectively[63]. The in-situ ATR-SEIRAS spectra as a function of time also display the same reaction intermediates over Co/Co₃O₄/CoB surface (Supplementary Fig. 45). Compared to Co/Co₃O₄ (Fig. 5b), a large number of important reactive intermediates including *NH and *NH₂ can be observed over Co/Co₃O₄/CoB surface at less negative cathodic potentials, suggesting much enhanced NORR reaction kinetics on Co/ Co₃O₄/CoB. On the other hand, a larger amount of *NH and *NH₂ intermediates were observed accumulating on Co₃O₄ surface compared with Co/Co₃O₄ and Co/Co₃O₄/CoB, implying the weaker ability of supplying *H on Co₃O₄ (Supplementary Fig. 46). To further confirm the key reaction intermediates, the DEMS measurement was

performed over Co/Co$_3$O$_4$/CoB at −0.5 V vs. RHE. As displayed in Fig. 5c and Supplementary Fig. 47, it can be observed that NO decreases with a decrease of NOH and HNOH intermediates, and meanwhile NH, NH$_2$, and NH$_3$ increase. These intermediates observed in DEMS align well with the results obtained from in-situ ATR-SEIRAS measurements. Therefore, the NORR pathway taking place on the Co/Co$_3$O$_4$/CoB surface is speculated to follow *NO → *HNO→ *NHOH → *NH → *NH$_2$ → *NH$_3$ → NH$_3$. Additionally, the signal of NH$_3$ in the DEMS measurement is much stronger than that of H$_2$, N$_2$, NH$_2$OH, and N$_2$O, explaining the high FE$_{NH3}$ in NORR over Co/Co$_3$O$_4$/CoB. Figure 5d compares the reaction energy diagram of NORR over Co/Co$_3$O$_4$/CoB and Co/Co$_3$O$_4$ surface (the structural models of intermediates are shown in Supplementary Figs. 48 and 49). Compared with *NOH, the first protonation of NO is more energetically favorable to form *HNO (Supplementary Figs. 50 and 51), which is the potential-determining step (PDS) of NORR on Co/Co$_3$O$_4$/CoB and Co/Co$_3$O$_4$. Clearly, Co/Co$_3$O$_4$/CoB shows a lower reaction energy barrier (0.81 eV) in PDS than that (1.11 eV) of Co/Co$_3$O$_4$, owing to the accelerated protonation process of NO over Co/Co$_3$O$_4$/CoB. The second proton preferentially reacts with *HNO to generate *HNOH with a ΔG of −1.88 eV, rather than *N with a ΔG of −0.82 eV. Finally, three successive protonation processes occur until NH$_3$ is formed. Besides, the reaction energy of NORR was also calculated over Co$_3$O$_4$, Co, and CoB (Supplementary Figs. 52–55 and Table 7). The reaction energy of the potential-determining step (*NO to *HNO) is 1.25, 2.81 and 1.6 eV over Co$_3$O$_4$, Co, and CoB, respectively, much higher than that over Co/Co$_3$O$_4$/CoB. This result evidences that the first step of NO hydrogenation can be notably boosted over the ternary Co/Co$_3$O$_4$/CoB heterostructure.

## Discussion

In summary, we have designed and prepared a ternary Co/Co$_3$O$_4$/CoB heterostructure through a thermal solid-state reduction reaction of Co$_3$O$_4$ by NaBH$_4$. The prepared Co/Co$_3$O$_4$/CoB displayed a high FE$_{NH3}$ of 98.80% with NH$_3$ yield rate of 462.18 μmol cm$^{-2}$ h$^{-1}$ (2.31 mol h$^{-1}$ g$_{cat}^{-1}$) and long-time stability at a low cathodic potential of −0.5 V vs. RHE, outperforming most of the reported NORR electrocatalysts in the literature. The extraordinary NORR performance of Co/Co$_3$O$_4$/CoB was resulted from the unique charge and mass transfer within the Co/Co$_3$O$_4$/CoB heterostructure (Fig. 5e). The enhanced electron transfer in Co/Co$_3$O$_4$/CoB yielded electron-deficient Co and electron-rich Co$_3$O$_4$. The electron-deficient Co sites could effectively boost H$_2$O dissociation to generate *H while the electron-rich low coordination Co$_3$O$_4$ sites promoted NO adsorption. The *H formed on electron-deficient Co site was more favorable to transfer to electron-rich Co$_3$O$_4$ site adsorbed with NO, facilitating the selective hydrogenation of NO. Thanks to the enhanced charge and proton transfer, the energy barrier of the potential-determining step from *NO to *HNO in NORR over Co/Co$_3$O$_4$/CoB was significantly reduced. Moreover, the introduction of CoB in Co/Co$_3$O$_4$/CoB could also facilitate the adsorption of *NH and *NH$_2$ intermediates. All of which greatly promoted electrochemical NORR. This study paves the way for designing and developing highly efficient electrocatalysts for reducing NO to NH$_3$.

## Methods
### Reagents and materials
Carbon paper (99.99%) and Ketjen carbon were purchased from Suzhou Sinero Technology Co., Ltd. Cobalt(II) acetate tetrahydrate (Co(CH$_3$COO)$_2$·4H$_2$O, ≥99%), polyvinyl pyrrolidone (PVP, ≥99%), sodium borohydride (NaBH$_4$, ≥98.0%), chloride (NH$_4$Cl, 99.99%), ethanol (C$_2$H$_5$OH, ≥99.5%), were purchased from Shanghai Titan Scientific Co., Ltd. Hydroxylammonium chlorideammonium (NH$_2$OH·HCl, ≥99%), dimethyl Sulfoxide-D6 (C$_2$D$_6$OS, 99.8%), sodium hydroxide (NaOH, ≥98%), trisodium citrate (C$_6$H$_5$Na$_3$O$_7$, 98%), sodium nitroprusside dihydrate (C$_5$H$_4$FeN$_6$Na$_2$O$_3$·H$_2$O, ≥99.98%), sulfuric acid

(H$_2$SO$_4$, >95%), hydrogen peroxide (H$_2$O$_2$, 30%), and phosphate buffer solution (PBS, 10X, pH = 7.3 ± 0.1) were purchased from Shanghai Adamas Reagents Co., Ltd. Sodium hypochlorite solution (NaClO, available chlorine 4.0%), salicylic acid (C$_7$H$_6$O$_3$, 99%) were purchased from Macklin Biochemical Technology Co., Ltd. Deionized water (18.25 MΩ cm resistivity) was obtained via an ultrapure water equipment in laboratory. All the reagents are analytical-grade and directly used without further purification.

### Synthesis of Co$_3$O$_4$ nanosheets
The Co$_3$O$_4$ nanosheets were prepared by a hydrothermal method as follows: Co(CH$_3$COO)$_2$.4H$_2$O (0.75 g) and PVP (2.4 g) were dissolved in 70 mL methanol under stirring for 30 min. The mixture was transferred into a 100 mL Teflon-lined stainless-steel autoclave that was sealed and heated to 190 °C for 12 h. After cooling to 25 °C, the products were harvested by centrifugation, washed three times with ethanol, dried at 60 °C, and calcined at 350 °C for 5 h in air[64].

### Synthesis of Co/Co$_3$O$_4$/CoB and Co/Co$_3$O$_4$
In a typical synthesis of Co/Co$_3$O$_4$/CoB, Co$_3$O$_4$ and NaBH$_4$ were uniformly mixed in a mortar at a mass ratio of 1:1.5. The mixture was then placed in a horizontal quartz tube and heated to 500 °C at a ramp rate of 2 °C min$^{-1}$ under an Ar atmosphere, which was maintained at 500 °C for 4 h. The resulting products were washed extensively by an ethanol/deionized water mixture and collected by filtration. Finally, the washed Co/Co$_3$O$_4$/CoB was dried at 80 °C for 12 h in a vacuum oven. For preparing Co/Co$_3$O$_4$, the mass ratio of Co$_3$O$_4$ and NaBH$_4$ was adjusted to 1.5:1, while other preparation parameters were kept unchanged.

### Electrochemical measurements
Electrochemical measurements were performed on a CHI660E electrochemical workstation in an H-type electrochemical cell (Supplementary Fig. 56) separated by a Nafion 117 membrane (10*10 cm, thickness was 183 μm, Dupont) in a three-electrode configuration at 25 °C. We chose 0.1 M PBS buffer solution (pH = 7.3 ± 0.1) to maintain the pH of the electrolyte during NORR. The membrane was sequentially treated in a H$_2$O$_2$ (5 wt.%) aqueous solution at 80 °C for 1 h, then in a 0.5 M H$_2$SO$_4$ solution at 80 °C for 2 h and finally in deionized water at 25 °C for 6 h. The catalyst coated on carbon paper (CP,1 cm$^2$) was used as the working electrode, a platinum plate (1 cm$^2$) was used as the counter electrode, and an Ag/AgCl (saturated 3.5 M KCl aqueous solution) electrode was used as the reference electrode. The Ag/AgCl reference electrode was calibrated in H$_2$-saturated 0.1 M PBS using a symmetric Pt electrode system.

To prepare the working electrode, 2.5 mg of electrocatalyst and 2.5 mg of Ketjen carbon were ultrasonically dispersed in 300 μL ethanol, 150 μL ultrapure water, and 25 μL Nafion solution (5 wt.%, Du Pont) to form a homogeneous catalyst ink followed by dropping 38 μL of the catalyst ink onto a piece of CP (1 cm$^2$) that was dried at 25 °C, the catalyst mass loading was 0.2 mg cm$^{-2}$. Before all electrochemical tests, 30 min of high-purity Ar gas (99.999%) and 30 min of NO/Ar gas (10 vol.%) saturated the electrolyte to exclude air in the reaction system.

Linear sweep voltammetry (LSV) was performed at a scanning rate of 5 mV/s prior to 50 cycles of cyclic voltammetry at a scan rate of 50 mV/s to obtain a stable curve with 90% iR-correction. The non-iR corrected data for all catalysts are provided in the Supplementary Information. The electrochemical impedance spectroscopy (EIS) was obtained without iR-correction in the frequency range from 0.01 Hz to 100 kHz upon an AC voltage amplitude of 5 mV at an open-circuit potential under 25 °C. The chronoamperometry was operated to evaluate the stability under continuous stirring (1600 rpm) at different current densities. All potentials in this study were converted to the

reversible hydrogen electrode (RHE) scale according to the following equation:

$$E(V \text{ vs.RHE}) = E(V \text{ vs.Ag/AgCl}) + 0.198\,V + 0.059 \times pH \quad (1)$$

Calculation of Faradaic efficiency (FE) and yield rate (Y)

$$FE_i(\%) = \frac{Q_i}{Q_{total}} \times 100\% = \frac{c_i \times V \times n \times F}{M_i \times Q_{total}} \times 100\% \quad (2)$$

$$Y_i(\mu mol\,h^{-1}\,cm^{-2}) = \frac{V \times c_i}{M_i \times t \times s} \quad (3)$$

Where $Q_i$ is the charge of product $i$, $Q_{total}$ is the total charge, $c_i$ is the concentration of products ($\mu g\,mL^{-1}$), V is the volume of electrolyte in the cathode compartment (25 mL), n: the number of electron transfer in products; F: Faraday constant (96485 C $mol^{-1}$); $M_i$: the molar mass of products (g $mol^{-1}$); s: the area of the electrode (1 $cm^2$); t: the reaction time (1 h).

## ECSA analysis
Electrochemical active surface area (ECSA) was evaluated through cyclic voltammetry (CV) measurements conducted within the non-Faradaic potential region in 0.1 M PBS electrolyte, where iR drop is negligible. The electrochemical double-layer capacitance ($C_{dl}$) was determined from the linear slope of capacitive current ($\Delta j = 0.5 \times |j_{charge} - j_{discharge}|$) plotted against scan rate (20–120 mV $s^{-1}$). The ECSA of the catalyst was determined by normalizing the double-layer capacitance ($C_{dl}$) against the standard specific capacitance ($C_s$) for smooth metal surfaces in 1.0 M netural electrolyte, following the relation: ECSA = $C_{dl}/C_s$.

## Quantification of NH$_3$
The yield rate of NH$_3$ in NORR was quantitatively determined by the indophenol blue method and NMR. For the indophenol blue method, the concentration-absorbance curves were calibrated using a standard NH$_3$ solution with a series of concentrations. The fitting curve (y = 0.4149x − 0.0264, $R^2$ = 0.9999) shows a good linear relationship between absorbance and NH$_3$ concentration. For NMR measurement, after NORR, the electrolyte was taken out, whose pH was adjusted to 3 using a 0.5 M HCl aqueous solution, and then sent for quantification by $^1$H NMR (600 MHz) with internal standard of maleic acid (3.5 M). The number of scans was 128 for all NMR measurements. The fitting curve (y = 2.7375x − 1.9166, $R^2$ = 0.9993) shows a good linear relationship.

## $^{15}$N isotope labeling experiment
The $^{15}$N isotope labeling experiment proceeded on the $^1$H NMR spectroscopy (600 MHz) to identify the nitrogen source for NORR. Replace $^{14}$NO with $^{15}$NO to conduct the NORR experiment at −0.5 V vs. RHE. After the experiment, take 2 mL of the catholyte were acidized by H$_2$SO$_4$ (2 mL, 1 M). 60 μL of the above solutions were respectively added to a NMR tube and mixed with maleic acid aqueous solution (20 μL, 3.6 mM), H$_2$SO$_4$ aqueous solution (20 μL, 4 M) and DMSO-d$_6$ (500 μL), and then analyzed by $^1$H NMR measurements.

## Quantification of hydroxylamine (NH$_2$OH)
To quantify NH$_2$OH, the following literature procedures were executed: 1.0 mL of the catholyte (or 0.1 mM-0.5 mM standard NH$_2$OH solution) was added 1.0 mL PBS buffer (pH = 7.3 ± 0.1) and 1.0 mL 1% 8-hydroxylquinoline. Under vigorous shaking, 1.0 mL 0.1 M Na$_2$CO$_3$ was added and the mixture was heated at 100 °C for 1 min. In the presence of NH$_2$OH, the solution would turn from light yellow to blue-green, showing an absorption peak at ~705 nm. Plotting the peak absorbance with NH$_2$OH concentration yielded a calibration curve.

## Zn-NO battery
A Co/Co$_3$O$_4$/CoB coated on carbon paper was employed as the cathode to perform the NORR in a cathodic electrolyte (0.1 M PBS). A polished Zn plate was applied as the anode in an anodic electrolyte (1 M KOH), and a Nafion 117 membrane was used to separate the cathode from the anode. The Zn-NO battery was assessed on a CHI660E electrochemical workstation under an ambient atmosphere at 25 °C.

## Characterizations
The morphological information was characterized by atomic force microscopy (AFM, Park NX10), field emission scanning electron microscopy (SEM, Zeiss-sigma-300), and transmission electron microscopy (TEM, JEOL JEM-1400). High-resolution transmission electron microscopy (HRTEM) and aberration-corrected high-angle annular dark-field scanning transmission electron microscopy (HAADF-STEM) were conducted on a JEOL JEM 2100 F and JEM-ARM 300 F Grand ARM. X-ray diffraction (XRD) patterns were recorded on a Bruker D8 Advance Diffractometer (Cu-Kα radiation: λ = 0.15406 nm). The surface valence states were studied by X-ray photoelectron spectroscopy (XPS, PHI-5300) with Mg Kα radiation (hν = 1486.6 eV). The X-ray absorption near edge structure (XANES) and extended X-ray absorption fine structure (EXAFS) spectra at the Co K-edge were collected at the HXMA beamline of the Canadian Light Source (CLS) using Fluorescence mode and Si (111) monochromator. The samples were pressed into wafers for collecting the data at 25 °C. The EXAFS raw data were background subtracted, normalized and Fourier transformed by standard procedures using the ATHENA program. The acquired EXAFS data were processed according to the standard procedures using the ATHENA module implemented in the IFEFFIT software packages. The $k^3$-weighted EXAFS spectra were obtained by subtracting the post-edge background from the overall absorption and then normalizing for the edge-jump step. The ultraviolet-visible (UV–vis) absorbance spectra were measured on an Agilent S3 Cary 5000. A Bruker 600 M NMR instrument with water suppression was used to record the $^1$H NMR spectra. The Brunauer-Emmett-Teller (BET) surface area was measured on a U.S. Quantachrome ASAP 2020 M at 77 K.

## NO adsorption breakthrough measurements
The NO adsorption breakthrough measurements were carried out on a dynamic sorption analyzer (mixSorb S) equipped with a thermal conductivity detector (TCD) combined with a mass spectrometer (OMNISTARTM). Firstly, 40 mg of catalyst was pre-treated in a high purity He atmosphere at a flow rate of 30 mL/min for 1 h. Afterwards, the sample was exposed to NO gas (0.02 vol.% NO balanced by He) at a total gas flow rate of 20 mL/min at 25 °C accompanied by recording the TCD signal and the MS signal of NO at m/z = 30 until the outlet mass signal of NO achieved saturation.

## In-situ attenuated total reflection-surface enhanced infrared absorption spectroscopy (ATR-SEIRAS) measurements
In-situ ATR-SEIRAS spectra were recorded on an INVENIO-R FTIR spectrometer (Bruker) equipped with a liquid nitrogen-cooled mercury cadmium telluride (MCT) detector (Supplementary Fig. 57). The catalyst ink was prepared by mixing 5 mg of catalyst, 300 μL ethanol, 150 μL ultrapure water, and 25 μL Nafion solution under sonication for 30 min. Next, 40 μL catalyst ink was slowly dropped onto a face-angled Si crystal to prepare the working electrode of the custom-made spectroelectrochemical cell fixed on the ATR accessory. Ag/AgCl electrode was employed as the reference electrode and a Pt wire was used as the counter electrode. Electrolyte was added in advance with continuous Ar flow at a flow rate of 50 mL/min for 30 minutes to remove interference of H$_2$O and O$_2$ prior to ATR-SEIRAS measurement. The background was measured at open circuit potential. Subsequently, 10% NO/Ar gas was injected and the absorption spectra were

recorded at different applied potentials from OCP to −0.7 V vs. RHE. Afterwards, spectrum was collected at −0.5 V vs. RHE every 5 min for 60 min.

## Online differential electrochemical mass spectroscopy (DEMS) measurements

In-situ DEMS measurement was conducted on Linglu (Supplementary Fig. 58). A Teflon layer was applied to separate the electrolyte from the vacuum system. The vacuum system, comprising of two dry pumps and one turbo pump, kept the vacuum degree below $10^{-7}$ Pa. The catalyst ink was prepared by mixing 3 mg of catalyst, 460 µL ethanol, 500 µL ultra-pure water, and 40 µL Nafion solution under sonication for 30 min. Next, 40 µL catalyst ink was slowly dropped onto gold film to prepare the working electrode. Ag/AgCl electrode was employed as the reference electrode and a Pt wire was used as the counter electrode. Before the DEMS measurement, NO/Ar was bubbled into the electrolyte until saturation. I-$t$ test for 100 s was used to conduct DEMS investigations.

## In-situ Raman spectroscopy

The in-situ Raman measurements were carried out jointly by an RXN1 Raman instrument (KAISER OPTICAL SYSTEM) and a CHI660E electrochemical workstation. A custom-made spectroelectrochemical cell was used as a reactor to enable the in-situ measurements. The obtained $Co/Co_3O_4/CoB$ catalyst, Ag/AgCl, and platinum wire served as the working electrode, reference electrode, and counter electrode, respectively. The working electrode was immersed in the electrolyte and positioned such that the electrode plane was perpendicular to the laser. In-situ Raman spectra were obtained while the electrodes were under potentiostatic control. The experiment was conducted for 200 s under each fixed potential.

## EPR measurements

5,5-dimethyl-1-pyrroline N-oxide (DMPO) was used to capture the unstable hydrogen radical to form the DMPO·H adduct to generate EPR spectrum. Briefly, 5 mL of electrolyte was mixed with 100 µL of DMPO and the mixture was deoxygenated by bubbling Ar. The constant current electrolysis was carried out for 10 min in the H-type cell under the protection of Ar. EPR measurement was performed on a JES X320, JEOL Co. spectrometer operated at a frequency near 9.5 GHz, sweep width of 200 G, and power of 20 mW. $Co/Co_3O_4/CoB$ fitting parameters: g = 2.0045, AN = 15.5 G, AH = 21.0 G, lwpp = 0.3. $Co/Co_3O_4$ fitting parameters: g = 2.0043, AN = 15.6 G, AH = 20.8 G, lwpp = 0.3.

## Density functional theory (DFT) calculations

Considering the limitations of computational power, we opted for a simplified mode, namely CoB supported neighboring Co and $Co_3O_4$ clusters, combininng with the morphology and three interfaces of $Co/Co_3O_4/CoB$ heterostructure. The choice of using crystal facets of Co (111), $Co_3O_4$ (311), and CoB (021) to model $Co/Co_3O_4/CoB$ heterostructure is primarily based on the XRD characterizations. These crystal facets exhibit the strongest diffraction in the XRD patterns and are also the thermodynamically stable facets. It is noted that the Co (111) and $Co_3O_4$ (311) clusters are taken from (111) facet of Co single crystal and (311) facet of $Co_3O_4$ single crystal, respectively.

All calculations were performed in the framework of the density functional theory with the projector augmented plane-wave method, as implemented in the Vienna ab initio simulation package (VASP)[65]. The generalized gradient approximation proposed by Perdew, Burke, and Ernzerhof was selected for the exchange-correlation potential[66]. The Grimme D3 correction used a coordination number dependent dispersion correction[67]. The cut-off energy for the plane wave was set to 450 eV. The energy criterion was set to $10^{-5}$ eV in the iterative solution of the Kohn-Sham equation. A vacuum layer of 15 Å was added perpendicular to the sheet to avoid artificial interaction between periodic images. The Brillouin zone integration was performed using a $2 \times 2 \times 1$ k-mesh[68]. All the structures were relaxed until the residual forces on the atoms had declined to less than 0.05 eV/Å. The maximum atomic force in $Co/Co_3O_4/CoB$ is 0.047, which meets the convergence requirement. In addition, we also evaluated the structural stability by using ab initio molecular dynamics (AIMD). The Brillouin zone was sampled using $1 \times 1 \times 1$ k-point grid. Self-consistent calculations were conducted with an energy convergence threshold of $10^{-5}$ eV. The AIMD was performed within the canonical (NVT) ensemble by Nosé-Hoover thermostats with a time step of 1.0 fs at a finite temperature of 300 K[69]. In the AIMD simulation up to 10 ps, the heterostructure of $Co/Co_3O_4/CoB$ was not destroyed, and the basic crystal structure was stable, proving the rationality of the structure (Supplementary Figs. 59–61 and Supplementary Data 2). Spin polarization was included in the calculations and the default setting of magnetic moment (MAGMOM= number of atoms of Co*1.0) was chosen for all calculations. The Gibbs free energy (ΔG) of reaction intermediates was calculated by the following:

$$\Delta G = \Delta E + \Delta E_{ZPE} - T\Delta S \qquad (4)$$

where ΔE is the adsorption energy. $\Delta E_{ZPE}$ and ΔS are the difference for the zero-point energy and entropy, respectively. The zero-point energy and entropy were calculated at the standard conditions corresponding to the pressure of 101325 Pa (~1 bar) of $H_2$ at the temperature of 298.15 K. The climbing image nudged elastic band (cNEB) method was used to search the reaction path and transition state, and the vibration frequency calculation was used to confirm it further[70].

## Data availability

Source data for all the figures and tables generated in this study are provided as a Source Data file. Source data are provided with this paper.

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

## Acknowledgements

The authors gratefully acknowledge financial support from the National Natural Science Foundation of China (22125604 and 22436003 to D.S.Z.), Shanghai Rising-Star Program (22QA1403700 to L.P.H.) and Science and Technology Commission of Shanghai Municipality (23230713700 to D.S.Z.), the City University of Hong Kong startup fund (9020003 to B.L.), ITF–RTH - Global STEM Professorship (9446006 to B.L.), and JC STEM lab of Advanced $CO_2$ Upcycling (9228005 to B.L.). X.Y.W. acknowledges the receipt of support from the CLSI Student Travel Support Program. XAFS spectra were tested at the HXMA beamline of the Canadian Light Source (CLS), a national research facility of the University of Saskatchewan, which is supported by the Canada Foundation for Innovation (CFI), the Natural Sciences and Engineering Research Council (NSERC), the National Research Council (NRC), and the Government of Saskatchewan. The authors thank Shenzhen HUA-SUAN Technology Co., Ltd for assistance on DFT calculations (https://huasuankeji.com).

## Author contributions

X.X.F. performed the experiments, analyzed the data, and wrote the first draft of the manuscript. Z.Y.T., Y.J.S., W.Q.Q., J.L.S., Z.L.W. and H.Y.D. helped with the measurements and analysis, discussed and revised the manuscript. X.Y.W. and Y.A.W. performed the XAFS measurements and analysis. L.P.H., B.L. and D.S.Z. supervised the project, manuscript preparation and discussion.

## Competing interests

The authors declare no competing interests.
