## [Transparent Peer Review file · Nature Communications]

Boosted charge and proton transfer over ternary Co/Co₃O₄/CoB for electrochemical nitric oxide reduction to ammonia

Corresponding Author: Professor Bin Liu

Version 0:

Reviewer comments:

Reviewer #1

(Remarks to the Author)

In this manuscript, Zhang and colleagues explored the structural advantages and electrochemical NORR applications of ternary Co/Co₃O₄/CoB. The ternary Co/Co₃O₄/CoB catalyst displays a high NH₃ Faradaic efficiency of 99.29% with an NH₃ yield of 462.18 μmol cm⁻² h⁻¹ at -0.5 V versus reversible hydrogen electrode, outperforming most of the reported NORR electrocatalysts to date. The super NORR performance can be attributed to the unique charge and mass transfer within the Co/Co₃O₄/CoB heterostructure with electron-deficient Co and electron-rich Co₃O₄. The CoB mediated electron-deficient Co sites can boost H₂O dissociation to generate *H, while the electron-rich low-coordination Co₃O₄ sites can promote NO adsorption and *H transfer. However, this paper fails to clearly describe electron transfer in ternary Co/Co₃O₄/CoB heterojunction, lacks a supported DFT calculation model, and provides experimental data without in-depth analysis, merely emphasizing the results without explaining their underlying reasons. Therefore, the current manuscript is not suitable for publication in Nature Communications.

Here are the detailed questions:

1. Explain how you can show that electrons are transferred from Co to CoB and from CoB to Co₃O₄ based on the XPS results for Co/Co₃O₄/CoB, rather than Co transferring electrons to CoB and Co₃O₄.
2. The authors should provide a clear explanation of the criteria for selecting Co and CoB clusters to model Co (111) and CoB (210).
3. Among the tested potentials, applied -0.5 V demonstrated the highest NH₃ Faradaic efficiency and NH₃ yield. Could the authors elaborate on the possible reasoning behind this observation? What are the underlying factors at this potential that contribute to the enhanced NORR performance?
4. The five NORR cycles test is not an indication of excellent stability of Co/Co₃O₄/CoB. The stability test should extend to more than 100 h.
5. The catalyst after the stability test should be characterized to support the structural stability of Co/Co₃O₄/CoB (e.g. XRD, XPS, atomic resolution HAADF-STEM image).
6. ¹⁵N Isotope labeling experiment should be performed to confirmed the source of as produced ammonia.
7. Fig. 4a does not include a calculation of the adsorption energy between the active site and the oxygen site in the NO molecule.
8. In Fig. 5e, the free energy of H₂O dissociation should represent the energy barrier of the transition state, rather than the total heat released in the final state.
9. The signals observed in the in situ FTIR in Figure S30 are not sufficiently robust to support the findings claimed by the authors. Additionally, performing DFT calculations on the NORR process of Co₃O₄ would provide valuable insights into the structure-activity relationship and enrich the overall analysis.
10. Please check the bond length of Co-B in Fig. 2e and provide a description of Fig. 3i.

Reviewer #2

(Remarks to the Author)

The authors designed polyethyleneimine (PEI) modified Au core Rh shell nanodendrites (Au@Rh-NDs) nano hybrids (Au@Rh-NDs/PEI) as a highly active electrocatalyst for NO₂ electroreduction to NH₃ (NO₂RR).

Detailed experimental studies were conducted to elucidate the origin for the improved catalytic NO₂RR reactivity. The presented results represent significant achievements in the field of NO₂RR electrocatalysis in terms of new synthetic method, fundamental understanding of the catalytic mechanism and impressive performances and will be attractive to broad audience. Overall, I recommend its publication after minor revision. Some specific comments are provided as follows

1. It is known that NO can be easily oxidized to NO₂ and generate NO₃⁻. NO₃⁻ has been widely reported that can be easily electroreduced to ammonia at a high Faradaic efficiency (>95%). Therefore, the authors are required to compare the energy efficiency between NO electroreduction, and "NO oxidation to NO₃⁻ coupled with NO₃⁻ reduction" pathways.
2. More recently reported NORR catalysts can be properly cited to make this paper state-of-the-art: ACS Energy Lett., 8 (2023) 1281-1288. Nano Lett., 23 (2023) 1735-1742. Nano Res., 16 (2023) 8737-8742.
3. The pH value of the electrolyte may be varied accompanied with ammonia production. How the catalytic stability can be maintained?
4. The structural information of Co/Co₃O₄/CoB after durability test can be provided.
5. The reaction energy, zero-point energy, and entropy corrections of each intermediate in the free energy calculations should be provided in detail (listed in tables of supporting information).

Reviewer #3

(Remarks to the Author)

In this manuscript, Dengsong Zhang and coworkers reported ternary Co/Co₃O₄/CoB as the integral electrocatalyst for nitric oxide reduction reaction (NORR), achieving a remarkable NH₃ production rate of 1462.18 μmol cm⁻² h⁻¹ at -0.5 V. However, from the aspect of catalyst design, electrocatalytic performance, NORR performance and mechanism investigation, there are absence of some innovative breakthroughs. More importantly, the author highlights the charge and proton transfer over ternary Co/Co₃O₄/CoB, while did not clearly identify its role in the DFT part. Moreover, the authors should also present the stability, yield rate with unit of mol h⁻¹ g_{cat}⁻² for the NORR and what is the role of pH of an electrolyte. Therefore, it is not recommended this manuscript for publication in Nature Communications.

Some specific comments;

1. In abstract, not specified the key results obtained in this work. In addition, NH₃ yield of 462.18 μmol cm⁻² h⁻¹ should be written as yield rate, not yield.
2. No novelty in the materials and also synthesis part. In NORR experiments, how about using alkaline media since most of reports available on alkaline media since ammonia produced as NH₄⁺ which is basic condition.
3. Please provide experimental details for catalyst activation.
4. In XPS, orbitals should be in italics. The interpretation of XAS and XPS is poor.
5. Zn-NO battery result, y axis should be V vs. Zn/Zn²⁺. Also, there is possibility for formation of ZnO or Zn(OH)₂ during electrochemical battery discharge cycle. Authors, not studied in depth of battery analyses.
6. Figure 31, the current density value is negative or positive?. Negative for discharge and positive for charging?.
7. In situ ATR, the surface reconstruction materials must be provided.
8. In the introduction, the importance of NO₂-RR should be emphasized, and recent studies on NORR for ammonia synthesis should be discussed.
9. It is suggested to include a comparison with other state-of-the-art catalysts in the discussion section.
10. Besides NH₃, it is essential to thoroughly examine the formation of other products.
11. To further support their findings, the authors should include in situ Raman spectra at different applied potentials.

Reviewer #4

(Remarks to the Author)

In this work, Fan et al. prepared a ternary heterostructured Co/Co₃O₄/CoB catalyst for electrochemical NO reduction. The catalyst exhibit superior performance over the reported ones due to the synergetic roles of these three phases with various activity towards the H₂O splitting, NO adsorption, as well as H⁺ transfer. The DFT calculations have been conducted to support the experimental results. Overall, this is an interesting work regarding the NO removal and catalyst design. However, to the best of my knowledge, the simulations seem to be not rigorous to support the results and hypothesis. I suggest major revision after addressing the following comments.

- 1) From the Fig.1d, the catalyst shows three boundary between Co(111), CoB(210), and Co₃O₄(311) surfaces. However, they applied a CoB supported (Co₃O₄-Co) cluster model to represent the catalyst. More specifically,
 - i) Why is this type of model applied? As the cluster did not contain any crystal information about the Co₃O₄ and Co(111) surfaces.
 - ii) How to ensure the stability of this model? Did the authors conduct AIMD to validate it? Based on the illustration in Supplementary Figure 6, the configuration seems to be not well relaxed.
 - iii) What are the oxidation states of the Co₃O₄ phase in this model? 2+ or 3+ ? How were the magnetizations determined in this model? Why DFT+U is not applied to the Co₃O₄?
- 2) Since three surfaces are observed in the experiments, obviously, the synergetic interplay of these three surfaces balances the RDS and promotes the NORR. The authors should conduct the DFT calculations on each of the surface to explore the rate limiting steps and investigate how synergetic mechanism works on the boundary. Assuming that the diffusion steps are not rate limiting steps, the three phases might play roles in different stages of NORR reaction network, similar to the most recent work of Wang et al (Small 2024, 20, 2311439).

Reviewer #5

(Remarks to the Author)

Version 1:

Reviewer comments:

Reviewer #1

(Remarks to the Author)

I have carefully read the author's response. If the other reviewers agree that the article is suitable for publication, I am also willing to support its acceptance. I appreciate the author's efforts in addressing the comments and improving the manuscript, and I trust the editorial team's judgment in making the final decision.

Reviewer #2

(Remarks to the Author)

The authors have positively responded all the comments raised by the reviewers. Therefore, I recommend the acceptance of this paper for publication.

Reviewer #3

(Remarks to the Author)

The revised version improved well.

Reviewer #4

(Remarks to the Author)

The authors have addressed all my comments.

Reviewer #5

(Remarks to the Author)

Point-by-point response to Reviewers' comments

Reviewer #1:

In this manuscript, Zhang and colleagues explored the structural advantages and electrochemical NORR applications of ternary Co/Co₃O₄/CoB. The ternary Co/Co₃O₄/CoB catalyst displays a high NH₃ Faradaic efficiency of 99.29% with an NH₃ yield of 462.18 μmol cm⁻² h⁻¹ at -0.5 V versus reversible hydrogen electrode, outperforming most of the reported NORR electrocatalysts to date. The super NORR performance can be attributed to the unique charge and mass transfer within the Co/Co₃O₄/CoB heterostructure with electron-deficient Co and electron-rich Co₃O₄. The CoB mediated electron-deficient Co sites can boost H₂O dissociation to generate *H, while the electron-rich low-coordination Co₃O₄ sites can promote NO adsorption and *H transfer. However, this paper fails to clearly describe electron transfer in ternary Co/Co₃O₄/CoB heterojunction, lacks a supported DFT calculation model, and provides experimental data without in-depth analysis, merely emphasizing the results without explaining their underlying reasons. Therefore, the current manuscript is not suitable for publication in Nature Communications. Here are the detailed questions:

Response: We thank the reviewer for the time and efforts spent in assessing our manuscript. Based on the reviewer's valuable comments and suggestions, we have conducted additional experiments and DFT simulations to clarify the concerns raised by the reviewer. Below please find our point-to-point response letter.

1. Explain how you can show that electrons are transferred from Co to CoB and from CoB to Co₃O₄ based on the XPS results for Co/Co₃O₄/CoB, rather than Co transferring electrons to CoB and Co₃O₄.

Response: We greatly appreciate the reviewer for the constructive suggestion. Actually, electrons can be possibly transferred from Co to CoB, CoB to Co₃O₄, and Co to Co₃O₄ in Co/Co₃O₄/CoB. The electron flow direction among Co, CoB and Co₃O₄ can be elucidated based on Bader charge and charge density difference analysis, and XPS results. Firstly, we supplemented Bader charge analysis of Co/Co₃O₄ and Co/Co₃O₄/CoB models, as seen in Fig. R1 and Table R1 (the revised Supplementary Figures 6,7 and Supplementary Table 1). In Co/Co₃O₄ model, 3.64 |e| is transferred from Co to Co₃O₄. In Co/Co₃O₄/CoB model, Co₃O₄ gets 2.23 |e| and 2.36 |e| from Co and CoB, respectively. Clearly, electrons can be transferred from both Co and CoB to Co₃O₄. Besides, Co/Co₃O₄/CoB shows significant charge redistribution at the interface between CoB and Co/Co₃O₄, with electron transfer from Co to CoB and from CoB to Co₃O₄ (Fig. R2 and Fig. 2c). In this study, we emphasized the improved electron transfer between CoB and Co as well as CoB and Co₃O₄ after introducing CoB, which achieved more electron loss of Co and more electron gain of Co₃O₄ as compared to Co/Co₃O₄. As evidenced in the XPS spectra of Co/Co₃O₄/CoB (Fig. R3 and Fig. 2b), the Co⁰ peak of Co slightly shifts to a higher binding energy (780.08 eV), while the Co³⁺ (780.68 eV) and Co²⁺ (782.98 eV) peaks of Co₃O₄ shift toward lower binding energies as compared to those of Co/Co₃O₄, indicating enhanced electron transfer in Co/Co₃O₄/CoB. Moreover, the average numbers of electron transfer per Co atom in Co/Co₃O₄/CoB and Co/Co₃O₄ models were analyzed. As shown in Table R1 (the revised Supplementary Table 1), Co loses more electron (-0.22 e vs. -0.13 e) while Co₃O₄ gains more electron (+0.51 e vs. +0.08 e) in Co/Co₃O₄/CoB as compared to Co/Co₃O₄, further indicating that the introduction of CoB into Co/Co₃O₄ can indeed promote electron transfer. We have supplemented the experimental results and added the corresponding discussion into the revised manuscript and Supplementary Information as follows:

“In Fig. 2b, the Co 2p_{3/2} XPS spectrum of Co/Co₃O₄ exhibits three peaks at binding energies of

779.7, 781.0 and 783.1 eV, which can be attributed to Co^0 , Co^{3+} and Co^{2+} , respectively⁴³⁻⁴⁵. For $\text{Co}/\text{Co}_3\text{O}_4/\text{CoB}$, the Co^0 peak slightly shifts to a higher binding energy (780.1 eV), while the Co^{3+} (780.7 eV) and Co^{2+} (782.2 eV) peaks shift toward lower binding energies as compared to those of $\text{Co}/\text{Co}_3\text{O}_4$. Because Co^0 and $\text{Co}^{3+}/\text{Co}^{2+}$ mainly exist in Co and Co_3O_4 , respectively, it can be inferred that the introduction of CoB leads to more electron loss in Co and more electron gain in Co_3O_4 . To clarify the electron transfer among different interfaces in $\text{Co}/\text{Co}_3\text{O}_4/\text{CoB}$ and $\text{Co}/\text{Co}_3\text{O}_4$, DFT calculations were conducted. Firstly, the Bader charge was analyzed over $\text{Co}/\text{Co}_3\text{O}_4$ and $\text{Co}/\text{Co}_3\text{O}_4/\text{CoB}$ models. For the $\text{Co}/\text{Co}_3\text{O}_4$ model, 3.64 |e| is transferred from Co to Co_3O_4 (Supplementary Fig. 6, Supplementary Table 1). In $\text{Co}/\text{Co}_3\text{O}_4/\text{CoB}$, Co_3O_4 gets 2.23 |e| and 2.36 |e| from Co and CoB , respectively (Supplementary Fig. 7 and Supplementary Table 1). The charge density difference analysis of $\text{Co}/\text{Co}_3\text{O}_4/\text{CoB}$ indicates a significant charge redistribution at the interface between CoB and $\text{Co}/\text{Co}_3\text{O}_4$ (Fig. 2c), with electron transfer from Co to CoB and from CoB to Co_3O_4 . Furthermore, the average numbers of electron transfer per Co atom in $\text{Co}/\text{Co}_3\text{O}_4/\text{CoB}$ and $\text{Co}/\text{Co}_3\text{O}_4$ were analyzed. As compared in Supplementary Table 1, Co loses more electrons (-0.22 e vs. -0.13 e) and Co_3O_4 gains more electrons (+0.51 e vs. +0.08 e) in $\text{Co}/\text{Co}_3\text{O}_4/\text{CoB}$ as compared to $\text{Co}/\text{Co}_3\text{O}_4$, evidencing that CoB introduction can modulate the electronic structure of Co and Co_3O_4 in $\text{Co}/\text{Co}_3\text{O}_4/\text{CoB}$ ” (Page 4-5 in the revised manuscript).

Fig. R1. The Bader charge analysis for $\text{Co}/\text{Co}_3\text{O}_4$ (a) and $\text{Co}/\text{Co}_3\text{O}_4/\text{CoB}$ (b).

Fig. R2. The charge density difference analysis for $\text{Co}/\text{Co}_3\text{O}_4/\text{CoB}$.

Fig. R3. Co $2p_{3/2}$ XPS spectra of Co/Co₃O₄ and Co/Co₃O₄/CoB.

Table R1. Quantitative analysis of Bader charge over Co/Co₃O₄/CoB and Co/Co₃O₄.

Catalyst model	Component	Total transferred electrons	The average number of electron transfer per Co atom
Co/Co ₃ O ₄ /CoB	CoB	-2.36	-0.04
	Co ₃ O ₄	4.59	0.51
	Co	-2.23	-0.22
Co/Co ₃ O ₄	Co ₃ O ₄	3.64	0.08
	Co	-3.64	-0.13

2. The authors should provide a clear explanation of the criteria for selecting Co and CoB clusters to model Co (111) and CoB (021).

Response: Thanks for your valuable suggestion. The Co/Co₃O₄/CoB heterostructure composes of large Co and small Co₃O₄ nanoparticles supported on CoB nanosheets (Figs. R4-R5, Fig. 1e, and Supplementary Figure 2). Such a ternary Co/Co₃O₄/CoB heterostructure contains three interfaces of Co-Co₃O₄, Co-CoB, and Co₃O₄-CoB. Considering the limitations of computational power, we opted for a simplified model, namely Co and Co₃O₄ clusters supported on CoB, which is a commonly adopted approach in DFT calculations in literature (Nat. Commun., 2024, 15, 8444; Nat. Catal., 2021, 4, 1032; Nat. Commun., 2019, 10, 1166). We have added the above discussion into the revised manuscript:

“Herein, by balancing the computation power and calculation accuracy, the Co/Co₃O₄/CoB was modeled by Co (111) and Co₃O₄(311) clusters supported on CoB (021), while Co/Co₃O₄ was modeled by Co (111) clusters supported on Co₃O₄ (311) (for details see DFT calculations in Method section).” (Page 5 in the revised manuscript)

Fig. R4. HAADF-STEM image and the corresponding EELS mappings of Co, O, and B elements in Co/Co₃O₄/CoB.

Fig. R5. The HRTEM image of Co/Co₃O₄/CoB.

3. Among the tested potentials, applied -0.5 V demonstrated the highest NH₃ Faradaic efficiency and NH₃ yield. Could the authors elaborate on the possible reasoning behind this observation? What are the underlying factors at this potential that contribute to the enhanced NORR performance?

Response: We thank the reviewer for raising this interesting question. The NORR process requires the participation of five electrons and five protons ($\text{NO} + 5\text{e}^- + 5\text{H}^+ \rightarrow \text{NH}_3 + \text{H}_2\text{O}$), which involves adsorption of NO followed by NO hydrogenation. Accordingly, the NH₃ Faradaic efficiency (FE_{NH_3}) and NH₃ yield rate of NORR are mainly dependent on NO activation and H₂O dissociation to supply *H. As shown in Fig. R6, the FE_{NH_3} and NH₃ yield rate increased with increasing the applied cathodic potential from -0.3 to -0.5 V vs. RHE. Further increase in the cathodic potential decreased the FE_{NH_3} and NH₃ yield rate due to the competing hydrogen evolution reaction (HER), a phenomenon commonly observed in literature (J. Am. Chem. Soc., 2023, 145, 6899; J. Am. Chem. Soc., 2024, 146, 1004). As shown in Table R2, the optimal operating potential (-0.5 V vs. RHE) of NORR for our Co/Co₃O₄/CoB catalyst is comparatively lower than that (-0.6/0.7 V vs. RHE) for most of the reported catalysts in the literature, which can be attributed to the unique charge and mass transfer within the Co/Co₃O₄/CoB heterostructure with electron-deficient Co and electron-rich Co₃O₄. The CoB mediated electron-deficient Co sites can boost H₂O dissociation to generate *H, while the electron-rich low-coordination Co₃O₄ sites can promote NO adsorption. The *H formed on electron-deficient Co site is more favorable to transfer to electron-rich Co₃O₄ site adsorbed with NO, facilitating the selective hydrogenation of NO. Thanks to the enhanced charge and proton transfer, the energy barrier of the potential-determining step from *NO to *HNO in NORR over Co/Co₃O₄/CoB is significantly reduced. We have added the above discussion into the revised manuscript:

“The ternary Co/Co₃O₄/CoB achieves a high FE_{NH₃} of 98.8% and NH₃ yield rate of 462.18 μmol h⁻¹ cm⁻² as well as long-time stability at a low cathodic potential of -0.5 V vs. RHE, superior to most of the reported NORR electrocatalysts (Fig. 3f and Supplementary Table 4).” (Page 7 in the revised manuscript).

“The extraordinary NORR performance of Co/Co₃O₄/CoB was resulted from the unique charge and mass transfer within the Co/Co₃O₄/CoB heterostructure (Fig. 5e). The enhanced electron transfer in Co/Co₃O₄/CoB yielded electron-deficient Co and electron-rich Co₃O₄. The electron-deficient Co sites could effectively boost H₂O dissociation to generate *H while the electron-rich low coordination Co₃O₄ sites promoted NO adsorption. The *H formed on electron-deficient Co site was more favorable to transfer to electron-rich Co₃O₄ site adsorbed with NO, facilitating the selective hydrogenation of NO. Thanks to the enhanced charge and proton transfer, the energy barrier of the potential-determining step from *NO to *HNO in NORR over Co/Co₃O₄/CoB was significantly reduced. Moreover, the introduction of CoB in Co/Co₃O₄/CoB could also facilitate the adsorption of *NH and *NH₂ intermediates. All of which greatly promoted electrochemical NORR.” (Page 12-13 in the revised manuscript).

Fig. R6. FE_{NH₃} and NH₃ yield rate over Co/Co₃O₄/CoB at various cathodic potentials.

Table R2. Comparison of NH₃ yield rate and FE_{NH₃} of Co/Co₃O₄/CoB with the reported NORR electrocatalysts.

Catalyst	NH ₃ yield rate (μmol·h ⁻¹ ·cm ⁻²)	FE _{NH₃} (%)	Stability time (h)	V vs. RHE	References
Co/Co ₃ O ₄ /CoB	462.18	98.80	~100	-0.5	This work
hcp-Co	439.50	72.58	6	-0.6	R1
Nb-SA/BNC	295.2	48.14	56	-0.9	R2
Cu (111)	187.5	93.19	5	-0.59	R3
a-B _{2.6} C@TiO ₂ /Ti	216.39	87.6	12	-0.9	R4
Fe ₁ /MoS _{2-x}	288.2	82.5	15	-0.6	R5
Sb ₁ /a-MoO ₃	273.5	91.7	30	-0.6	R6
Cu ₁ /MoS ₂	337.5	90.6	20	-0.6	R7
NiO/TM	125.29	90	12	-0.6	R8
CoB/Co@C	315.4	~ 85	10	-0.6	R9
Sb ₂ S ₃	168.6	93.7	20	-0.7	R10

References

- R1. Wang, D. et al. Hexagonal cobalt nanosheets for high-performance electrocatalytic NO reduction to NH₃. *J. Am. Chem. Soc.* **145**, 6899-6904 (2023).
- R2. Peng, X. et al. Ambient electrosynthesis of ammonia with efficient denitration. *Nano Energy*. **78**, 105321 (2020).
- R3. Xiao, L. et al. Identification of Cu(111) as superior active sites for electrocatalytic NO reduction to NH₃ with high single-pass conversion efficiency. *Angew. Chem. Int. Ed.* **63**, e202319135 (2024).
- R4. Liang, J. et al. Amorphous boron carbide on titanium dioxide nanobelt arrays for high-efficiency electrocatalytic NO reduction to NH₃. *Angew. Chem. Int. Ed.* **61**, e202202087 (2022).
- R5. Chen, K. et al. Atomically Fe-doped MoS_{2-x} with Fe-Mo dual sites for efficient electrocatalytic NO reduction to NH₃. *Appl. Catal. B: Environ.* **324**, 122241 (2023).
- R6. Chen, K. et al. p-Block antimony single-atom catalysts for nitric oxide electroreduction to ammonia. *ACS Energy Lett.* **8**, 1281-1288 (2023).
- R7. Chen, K., Zhang, G., Li, X., Zhao, X. & Chu, K. Electrochemical NO reduction to NH₃ on Cu single atom catalyst. *Nano Res.* **16**, 5857-5863 (2023).
- R8. Liu, P. et al. High-performance NH₃ production via NO electroreduction over a NiO nanosheet array. *Chem. Commun.* **57**, 13562-13565 (2021).
- R9. Wu, B. et al. Boron-modulated electronic-configuration tuning of cobalt for enhanced nitric oxide fixation to ammonia. *Nano Lett.* **23**, 7120-7128 (2023).
- R10. Chen, K., Zhang, Y., Du, W., Guo, Y. & Chu, K. Atomically isolated and unsaturated Sb sites created on Sb₂S₃ for highly selective NO electroreduction to NH₃. *Inorg. Chem. Front.* **10**, 2708-2715 (2023).

4. The five NORR cycles test is not an indication of excellent stability of Co/Co₃O₄/CoB. The stability test should extend to more than 100 h.

Response: We thank the reviewer for the valuable suggestion. Based on which, we have performed the stability test again. As displayed in Fig. R7 (Supplementary Fig. 20), the Co/Co₃O₄/CoB catalyst shows a good current stability for ~100 h (the electrolyte was updated every 24 hours). We have updated the results and discussion in the revised manuscript:

“Moreover, this catalyst exhibits good current stability for ~100 h at -0.5 V vs. RHE (Supplementary Fig. 20).” (Page 7 in the revised manuscript).

Fig. R7. The long-term chronoamperometric curve recorded over Co/Co₃O₄/CoB at -0.5 V vs. RHE (The electrolyte was updated every 24 hours. Blue arrows indicate the renewal of fresh electrolyte).

5. The catalyst after the stability test should be characterized to support the structural stability of Co/Co₃O₄/CoB (e.g. XRD, XPS, atomic resolution HAADF-STEM image).

Response: Thanks for your valuable suggestion. Accordingly, we have supplemented the XRD pattern and HRTEM image of Co/Co₃O₄/CoB after stability test. As shown in Fig. R8 (the revised Supplementary Figure 21), no changes are observed on the two diffraction peaks between fresh and used Co/Co₃O₄/CoB catalyst, indicating no phase change during the NORR. As displayed in the HRTEM image in Fig. R9 (the revised Supplementary Figure 22), the used catalyst still shows the ternary heterostructure of Co/Co₃O₄/CoB. Additionally, we further investigated the XPS spectrum of Co/Co₃O₄/CoB after the stability test. As shown in Fig. R10 (the revised Supplementary Figure 23), the used Co/Co₃O₄/CoB sample shows the similar characteristic peaks of Co⁰, Co²⁺, and Co³⁺ as compared to the fresh one. These results indicate a good structural stability of Co/Co₃O₄/CoB during NORR. We have added these data and discussion into the revised manuscript and Supplementary Information:

“Co/Co₃O₄/CoB shows a good structural stability during the NORR based on the XRD, HRTEM and XPS characterizations (Supplementary Figs. 21-23).” (Page 7 in the revised manuscript).

Fig. R8. XRD patterns of Co/Co₃O₄/CoB before and after NORR durability test.

Fig. R9. The HRTEM image of Co/Co₃O₄/CoB after NORR durability test.

Fig. R10. Co 2p_{3/2} XPS spectra of Co/Co₃O₄/CoB before and after NORR durability test.

6. ¹⁵N Isotope labeling experiment should be performed to confirm the source of as produced ammonia.

Response: We thank the reviewer for the valuable suggestion. Accordingly, we have used ¹⁵N labeled ¹⁵NO to carry out the NORR over Co/Co₃O₄/CoB. As displayed in Fig. R11, the NMR spectra show the predominant characteristic double peaks with a separation of 72 Hz, corresponding to ¹⁵NH₄⁺. Combined with a series of control experiments (Fig. R12), it can be proved that NO is the sole nitrogen source for NH₃ production (Nature, 2022, 609, 71; Nat. Catal., 2023, 6, 402). The corresponding results and discussion have been added into the revised manuscript as:

“Using isotopic ¹⁵NO as the reactant, the characteristic double peaks (inset in Fig. 3d) appear with a separation of 72 Hz, corresponding to ¹⁵NH₄⁺, indicating that ¹⁵NO is the sole nitrogen source of NH₃ production.” (Page 7 in the revised manuscript).

Fig. R11. The ¹H nuclear magnetic resonance spectra to quantify NH₄⁺.

Fig. R12. NORR performance over $\text{Co/Co}_3\text{O}_4/\text{CoB}$ and carbon paper (CP) under different conditions (O/C represents open circuit condition). The inset shows the ^1H nuclear magnetic resonance spectra to quantify NH_4^+ .

7. Fig. 4a does not include a calculation of the adsorption energy between the active site and the oxygen site in the NO molecule.

Response: Thanks for your valuable comment. Based on which, we have supplemented the adsorption energy calculation of NO with O end on active site (Figs. R13-R14, Supplementary Figures 32-34). As shown in Fig. R13, the adsorption energy of NO with N end is obviously more negative than that with O end on various active sites, indicating that the adsorption of NO with N end is more favorable. The corresponding data and discussion have been added into the revised manuscript as follows:

“The NO adsorption energies with N-end/O-end on the Co sites of Co (111), Co_3O_4 (311), and CoB (021) and the B sites of CoB (021) were computed by DFT (Fig. 4a and Supplementary Figs. 32-34). The results indicate that NO prefers to adsorb on Co sites of Co_3O_4 (311) in the form of N-end adsorption, having the most negative adsorption energy of -3.64 eV .” (Page 9 in the revised manuscript).

Fig. R13. The calculated NO adsorption energy of (a) O end and (a) N end on Co (111), CoB (021) and Co_3O_4 (311).

Fig. R14. Schematic diagrams showing the calculation models for NO adsorption energy of oxygen end on Co site of Co (a) and Co₃O₄ (b), as well as B and Co sites of CoB (c & d).

8. In Fig. 5e, the free energy of H₂O dissociation should represent the energy barrier of the transition state, rather than the total heat released in the final state.

Response: Thank you for pointing out this mistake. We have made the corresponding correction in the revised manuscript:

“The energy barrier for water transformation from the adsorbed state (H₂O*) to the transition state (TS) on Co/Co₃O₄/CoB and Co/Co₃O₄ are 0.50 eV and 0.66 eV, respectively (Fig. 4e, Supplementary Figs. 35 and 36, Supplementary Table 6). This result indicates that the dissociation of H₂O is more kinetically favorable on Co/Co₃O₄/CoB.” (Page 9 in the revised manuscript).

9. The signals observed in the in situ FTIR in Figure S30 are not sufficiently robust to support the findings claimed by the authors. Additionally, performing DFT calculations on the NORR process of Co₃O₄ would provide valuable insights into the structure-activity relationship and enrich the overall analysis.

Response: Thanks for your valuable comment and suggestion. First of all, we like to apologize for our unclear description about the *in-situ* FTIR data. As shown in Fig. R15, a larger amount of *NH and *NH₂ intermediates are observed accumulating on Co₃O₄ surface compared with Co/Co₃O₄ and Co/Co₃O₄/CoB during NORR, which implies that the sluggish proton supply over Co₃O₄ would hamper the hydrogenation of intermediates to produce NH₃.

According to the reviewer’s suggestion, the NORR process over Co₃O₄ was calculated by DFT (Fig. R16), and it was found that the reaction energy of the potential-determining step (*NO to *HNO) over Co₃O₄ is obviously higher than that over Co/Co₃O₄ and Co/Co₃O₄/CoB, which corresponded well with the less *HNO on Co₃O₄ catalyst observed by *in-situ* FTIR (Fig. R15). This result further evidences that the first step of NO hydrogenation is notably boosted over the ternary Co/Co₃O₄/CoB catalyst. We have added the above results and discussion into the revised manuscript and Supplementary Information:

“On the other hand, a larger amount of *NH and *NH₂ intermediates were observed accumulating on Co₃O₄ surface compared with Co/Co₃O₄ and Co/Co₃O₄/CoB, implying the weaker ability of supplying *H on Co₃O₄ (Supplementary Fig. 45).” (Page 11 in the revised manuscript).

“Besides, the reaction energy of NORR was also calculated over Co₃O₄, Co, and CoB (Supplementary Figs. 51-54 and Table 7). The calculated reaction energy for the potential-determining

step (*NO to *HNO) is 1.25, 2.81 and 1.6 eV over Co₃O₄, Co, and CoB, respectively, all higher than that over Co/Co₃O₄/CoB. This result further evidences that the first step of NO hydrogenation is notably boosted over the ternary Co/Co₃O₄/CoB catalyst.” (Page 12 in the revised manuscript).

Fig. R15. *In-situ* ATR-SEIRAS spectra recorded over Co/Co₃O₄/CoB **a**, Co/Co₃O₄ **b**, and Co₃O₄ **c**, in 0.1 M PBS at the applied cathodic potential from OCP to -0.7 V vs. RHE.

Fig. R16. The calculated NORR Gibbs free energy diagrams over Co₃O₄, Co/Co₃O₄ and Co/Co₃O₄/CoB at 0 V vs. RHE.

10. Please check the bond length of Co-B in Fig. 2e and provide a description of Fig. 3i.

Response: Thanks for pointing out this mistake. As the signal intensity of the original sample is too poor, we have re-tested the sample and analyzed the data. As shown in Fig. R17 (the revised Fig. 2e), the signal intensity is much better and the bond length of Co-B is labeled at 1.6 Å, in line with the reported value in the literature (Adv. Mater., 2023, 35, 2209552).

Besides, we have added a relevant description of Figure 3i. As displayed in Fig. R18, the output discharge current density increases continuously from 0.5 to 8 mA cm⁻², and each step exhibits a stable discharging plateau, indicating that our cell has an excellent discharge capability. The corresponding data and discussion have been added into the revised manuscript as follows:

“To understand the local structure around Co atom in these samples, the EXAFS spectra were transformed into R-space. The R-space EXAFS spectra (Fig. 2e) display dominant peaks of Co-B (1.60 Å)⁴⁶, Co-Co (2.17Å)⁴⁷, and Co-O (1.44 Å)⁴⁸ coordination bonds in CoB, Co foil and Co₃O₄, respectively.” (Page 5 in the revised manuscript).

“The output discharge current density increases continuously from 0.5 to 8 mA cm⁻² and each step exhibits a stable discharging plateau, indicating that our cell has an excellent discharge capability (Supplementary Fig. 28).” (Page 7-8 in the revised manuscript).

Fig. R17. The FT-EXAFS spectra of Co/Co₃O₄, Co/Co₃O₄/CoB, Co foil, CoB, and Co₃O₄.

Fig. R18. Discharging tests at various current densities.

Reviewer #2:

The authors designed polyethyleneimine (PEI) modified Au core Rh shell nanodendrites (Au@Rh-NDs) nanohybrids (Au@Rh-NDs/PEI) as a highly active electrocatalyst for NO₂ electroreduction to NH₃ (NO₂RR). Detailed experimental studies were conducted to elucidate the origin for the improved catalytic NO₂RR reactivity. The presented results represent significant achievements in the field of NO₂RR electrocatalysis in terms of new synthetic method, fundamental understanding of the catalytic mechanism and impressive performances and will be attractive to broad audience. Overall, I recommend its publication after minor revision. Some specific comments are provided as follows

Response: We thank the reviewer for the time and efforts spent in assessing our manuscript. Based on the reviewer's valuable comments and suggestions, we have conducted additional experiments and DFT simulations to clarify the concerns raised by the reviewer. Below please find our point-to-point response letter.

1. It is known that NO can be easily oxidized to NO₂ and generate NO₃⁻. NO₃⁻ has been widely reported that can be easily electroreduced to ammonia at a high Faradaic efficiency (>95%). Therefore, the authors are required to compare the energy efficiency between NO electroreduction, and "NO oxidation to NO₃⁻ coupled with NO₃⁻ reduction" pathways.

Response: We thank the reviewer for the valuable suggestion. In a recent study, Wang et al. utilized plasma-treated commercial carbon cloth as an electrocatalyst, which exhibited 100% Faradaic efficiency of NO oxidation to NO₃⁻ at 1.13 V vs. RHE (Angew. Chem. Int. Ed., 2021, 60, 24605). The potential of NO₃⁻RR is generally at -0.65 V vs. RHE (Nat. Commun., 2023, 14, 8036). Based on the above data, coupling of NO oxidation with NO₃⁻RR requires a higher energy consumption compared to direct NORR that is generally at a cathodic potential of -0.6/0.7 V vs. RHE.

"The approach to couple NO oxidation with NO₃⁻ reduction is also considered as an alternative to NORR. But NORR to produce NH₃ generally requires lower energy consumption^{20, 21}." (Page 2 in the revised manuscript)

2. More recently reported NORR catalysts can be properly cited to make this paper state-of-the-art: ACS Energy Lett., 8 (2023) 1281-1288. Nano Lett., 23 (2023) 1735-1742. Nano Res., 16 (2023) 8737-8742.

Response: Thanks for your valuable suggestion. We have cited the relevant references in the introduction of the article, as follows:

"Constructing single atoms in amorphous metal oxides with oxygen vacancies to form metal-O moieties could accelerate the hydrogenation of NO to NH₃.²⁸⁻³⁰" (Page 2 in the revised manuscript).

28. Chen, K., Wang, G., Guo, Y., Ma, D. & Chu, K. Iridium single-atom catalyst for highly efficient NO electroreduction to NH₃. *Nano Res.* **16**, 8737-8742 (2023).

29. Chen, K., Wang, J., Zhang, H., Ma, D. & Chu, K. Self-Tandem Electrocatalytic NO Reduction to NH₃ on a W Single-Atom Catalyst. *Nano Lett.* **23**, 1735-1742 (2023).

30. Chen, K. et al. p-Block Antimony Single-Atom Catalysts for Nitric Oxide Electroreduction to Ammonia. *ACS Energy Lett.* **8**, 1281-1288 (2023)." (Page 19 in the revised manuscript).

3. The pH value of the electrolyte may be varied accompanied with ammonia production. How the catalytic stability can be maintained?

Response: We thank the reviewer for raising this question. The pH value of the electrolyte was maintained almost unchanged by 0.1 M PBS buffer solution, which is a common approach widely used in the literature (Nat. Energy, 2020, 5, 605; Angew. Chem. Int. Ed., 2023, 62, e202215782; ACS Energy Lett., 2020, 5, 3647; Chem Catal., 2022, 2, 622; Nano Res., 2022, 15, 4008-4013).

“We chose 0.1 M PBS buffer solution to maintain the pH of the electrolyte during NORR.” (Page 13 in the revised manuscript).

4. The structural information of Co/Co₃O₄/CoB after durability test can be provided.

Response: Thanks for your valuable suggestion. Accordingly, we have supplemented the XRD pattern and HRTEM image of Co/Co₃O₄/CoB after stability test. As shown in Fig. R19 (the revised Supplementary Figure 21), no changes are observed on the two diffraction peaks between fresh and used Co/Co₃O₄/CoB catalyst, indicating no phase change during the NORR. As displayed in the HRTEM image in Fig. R20 (the revised Supplementary Figure 22), the used catalyst still shows the ternary heterostructure of Co/Co₃O₄/CoB. Additionally, we further investigated the XPS spectra of Co/Co₃O₄/CoB after the stability test. As shown in Fig. R21 (the revised Supplementary Figure 23), the used Co/Co₃O₄/CoB sample shows the similar characteristic peaks of Co⁰, Co²⁺, and Co³⁺ as compared to the fresh one. These results indicate a good structural stability of Co/Co₃O₄/CoB during NORR. We have added these data and discussion into the revised manuscript and Supplementary Information:

“Co/Co₃O₄/CoB shows a good structural stability during the NORR based on the XRD, HRTEM and XPS characterizations (Supplementary Figs. 21-23).” (Page 7 in the revised manuscript).

Fig. R19. XRD patterns of Co/Co₃O₄/CoB before and after NORR durability test.

Fig. R20. The HRTEM image of Co/Co₃O₄/CoB after NORR durability test.

Fig. R21. Co $2p_{3/2}$ XPS spectra of Co/Co₃O₄/CoB before and after NORR durability test.

5. The reaction energy, zero-point energy, and entropy corrections of each intermediate in the free energy calculations should be provided in detail (listed in tables of supporting information).

Response: Thanks for your valuable suggestion. We have provided the calculation details in Table R3 and R4 in the revised Supplementary Information (the revised Supplementary Table 6 and 7).

Table R3. The reaction energy, zero-point energy, and entropy corrections in the free energy calculations for H₂O dissociation over Co/Co₃O₄/CoB and Co/Co₃O₄ surface.

		ΔE	TPZ	T ΔS	ΔG (eV)
Co/Co ₃ O ₄ /CoB	*H ₂ O	-1.21	0.71	0.13	-0.62
	*OH+*H	-1.19	0.54	0.08	-0.73
Co/Co ₃ O ₄	*H ₂ O	-1.08	0.66	0.06	-0.49
	*OH+*H	-0.70	0.46	0.02	-0.26

Table R4. The reaction energy, zero-point energy, and entropy corrections of each intermediate in the free energy calculations over Co/Co₃O₄/CoB, Co/Co₃O₄, CoB, Co₃O₄, and Co surface.

		ΔE	TPZ	$T\Delta S$	ΔG (eV)
Co/Co ₃ O ₄	*NO	-2.37	0.26	0.16	-2.27
	*NOH	-1.07	0.54	0.14	-0.67
	*HNO	-1.58	0.51	0.09	-1.16
	*N	-0.65	0.09	0.07	-0.63
	*HNOH	-2.29	0.84	0.14	-1.59
	*NH	-1.72	0.49	0.14	-1.36
	*NH ₂	-2.90	0.74	0.08	-2.24
	*NH ₃	-3.81	1.11	0.14	-2.84
Co/Co ₃ O ₄ /CoB	*NO	-2.80	0.24	0.10	-2.66
	*NOH	-1.73	0.53	0.10	-1.30
	*HNO	-2.24	0.53	0.13	-1.85
	*N	-0.90	0.12	0.03	-0.82
	*HNOH	-2.58	0.81	0.12	-1.88
	*NH	-1.84	0.40	0.07	-1.52
	*NH ₂	-2.95	0.70	0.07	-2.32
	*NH ₃	-3.75	1.11	0.13	-2.77
CoB	*NO	-2.05	0.24	0.11	-1.92
	*HNO	-0.79	0.52	0.05	-0.32
	*HNOH	-2.71	0.84	0.15	-2.02
	*NH	-1.53	0.34	0.03	-1.21
	*NH ₂	-3.35	0.75	0.13	-2.73
	*NH ₃	-3.91	1.14	0.09	-2.87
Co ₃ O ₄	*NO	-3.64	0.22	0.08	-3.51
	*HNO	-2.63	0.51	0.14	-2.26
	*HNOH	-4.19	0.82	0.09	-3.46
	*NH	-3.44	0.35	0.07	-3.16
	*NH ₂	-4.08	0.70	0.11	-3.49
	*NH ₃	-4.75	1.08	0.10	-3.78
Co	*NO	-2.68	0.21	0.04	-2.51
	*HNO	-1.12	0.53	0.12	-0.70
	*HNOH	-2.64	0.84	0.15	-1.95
	*NH	-1.92	0.37	0.09	-1.64
	*NH ₂	-2.74	0.69	0.11	-2.17
	*NH ₃	-3.60	1.07	0.10	-2.62

Reviewer #3:

In this manuscript, Dongsong Zhang and coworkers reported ternary Co/Co₃O₄/CoB as the integral electrocatalyst for nitric oxide reduction reaction (NORR), achieving a remarkable NH₃ production rate of 462.18 μmol cm⁻² h⁻¹ at -0.5 V. However, from the aspect of catalyst design, electrocatalytic performance, NORR performance and mechanism investigation, there is absence of some innovative breakthroughs. More importantly, the author highlights the charge and proton transfer over ternary Co/Co₃O₄/CoB, while did not clearly identify its role in the DFT part. Moreover, the authors should also present the stability, yield rate with unit of mol h⁻¹ g_{cat}⁻² for the NORR and what is the role of pH of an electrolyte. Therefore, it is not recommended this manuscript for publication in Nature Communications. Some specific comments:

Response: We thank the reviewer for the time and efforts spent in assessing our manuscript. Based on the reviewer's valuable comments and suggestions, we have conducted additional experiments and DFT simulations to clarify the concerns raised by the reviewer. Below please find our point-to-point response letter.

1. In abstract, not specified the key results obtained in this work. In addition, NH₃ yield of 462.18 μmol cm⁻² h⁻¹ should be written as yield rate, not yield.

Response: We thank the reviewer for the valuable comments. Currently, the FE_{NH₃} and NH₃ yield rate in electrochemical NORR are still impeded by the weak adsorption of NO and the sluggish proton supply. In this work, we attempt to improve NORR performance through simultaneously promoting charge and proton transfer. In order to achieve this goal, we tactfully designed and synthesized a ternary Co/Co₃O₄/CoB heterostructure by reducing Co₃O₄ with NaBH₄. The ternary Co/Co₃O₄/CoB catalyst exhibited an excellent electrochemical NORR performance, achieving an NH₃ yield rate of 462.18 μmol cm⁻² h⁻¹ and an FE_{NH₃} of 98.8% at -0.5 V vs. RHE, outperforming most of the reported NORR electrocatalysts to date. The super NORR performance is attributed to the unique charge and mass transfer within the Co/Co₃O₄/CoB heterostructure with electron-deficient Co and electron-rich Co₃O₄. The CoB mediated electron-deficient Co sites can boost H₂O dissociation to generate *H while the electron-rich low-coordination Co₃O₄ sites can promote NO adsorption. The *H formed on electron-deficient Co site is more favorable to transfer to electron-rich Co₃O₄ site adsorbed with NO, facilitating the selective hydrogenation of NO. The energy barrier for the potential-determining step from *NO to *HNO in NORR is thus greatly reduced over ternary Co/Co₃O₄/CoB. Based on the above discussion, the abstract has been reorganized as follows:

“Herein, we have tactfully designed and synthesized a ternary Co/Co₃O₄/CoB heterostructure that displays a high NH₃ Faradaic efficiency of 98.8% in NORR with an NH₃ yield rate of 462.18 μmol cm⁻² h⁻¹ (2.31 mol h⁻¹ g_{cat}⁻¹) at -0.5 V versus reversible hydrogen electrode, outperforming most of the reported NORR electrocatalysts to date. The outstanding NORR performance is attributed to the enhanced charge and proton transfer over the ternary Co/Co₃O₄/CoB heterostructure. The charge transfer between CoB and Co/Co₃O₄ yields electron-deficient Co and electron-rich Co₃O₄. The electron-deficient Co sites boost H₂O dissociation to generate *H while the electron-rich low-coordination Co₃O₄ sites promote NO adsorption. The *H formed on electron-deficient Co site is more favorable to transfer to electron-rich Co₃O₄ site adsorbed with NO, facilitating the selective hydrogenation of NO. Thereby, the energy barrier for the potential-determining step from *NO to *HNO in NORR over the ternary Co/Co₃O₄/CoB is greatly reduced.” (Page 1 in the revised manuscript).

In addition, we have changed “NH₃ yield” to “NH₃ yield rate” in the manuscript. As most literatures reported the NH₃ yield rate using the unit of $\mu\text{mol cm}^{-2} \text{h}^{-1}$, we also used this unit. According to the reviewer’s suggestion, we also calculated the NH₃ yield rate as $2.31 \text{ mol h}^{-1} \text{ g}_{\text{cat}}^{-1}$, which has been added into the revised manuscript. The figures have been revised as follows:

Fig. R23. Comparison of NH₃ yield rate and FE_{NH₃} over Co/Co₃O₄/CoB after 1 h of NO electrolysis at -0.6 V vs. RHE determined by UV-vis and NMR method.

Fig. R24. Ammonia production in the Zn-NO battery at different current densities.

2. No novelty in the materials and also synthesis part. In NORR experiments, how about using alkaline media since most of reports available on alkaline media since ammonia produced as NH_4^+ which is basic condition.

Response: We thank the reviewer for the comments. To the best of our knowledge, such an elaborately designed ternary Co/Co₃O₄/CoB heterostructure has not been reported in the literature for any catalytic reactions. Currently, the F_{NH_3} and NH_3 yield rate in electrochemical NORR are still impeded by the weak adsorption of NO and the sluggish proton supply. In order to address these challenges, we tactfully designed and synthesized a ternary Co/Co₃O₄/CoB heterostructure, in which the electron-deficient Co sites boosted H₂O dissociation to generate *H, while the electron-rich low-coordination Co₃O₄ sites promoted NO adsorption. Overall, we believe that our catalyst design concept is novel and the developed Co/Co₃O₄/CoB catalyst can notably improve NORR performance.

As for materials synthesis, the current method for preparing ternary heterostructure is very complicated. For example, the Co₃Mo₃N/Co₄N/Co ternary heterostructured catalyst was synthesized by growing ZIF-L nanowires on carbon cloth, followed by ion exchange to introduce molybdate ions, and finally through a high-temperature calcination (Angew. Chem. Int. Ed., 2024, 63, e202319239). In another example, the Ni/MoO₂@CN ternary heterostructure was synthesized via a solvothermal method, followed by H₂ reduction at high temperatures, and finally ultrasonication (Nano-Micro Lett., 2022, 14, 20). Our ternary Co/Co₃O₄/CoB heterostructure was synthesized by a simple one-step solid phase reduction method, namely by reducing Co₃O₄ using NaBH₄ in an Ar atmosphere at 500 °C for 4 h. In short, our synthesis method to construct ternary heterostructure is facile and simple.

The latest research of NORR shows that the F_{NH_3} is higher for the NORR conducted in neutral electrolyte as compared to that conducted in alkaline electrolyte (Nat. Commun., 2024, 15, 7243; Nat. Water, 2023, 1, 1068), and most reported NORR used neutral electrolyte (J. Am. Chem. Soc., 2023, 145, 6899; Angew. Chem. Int. Ed., 2024, e202319135; Angew. Chem. Int. Ed., 2023, 62, e202213351). In this study, we chose a neutral PBS buffer solution to maintain the electrolyte pH during NORR. Moreover, we also conducted NORR experiments under alkaline conditions. As shown in Fig. R25 (the revised Supplementary Fig. 18), the alkaline reaction environment yielded a poorer NORR performance. Additionally, alkaline electrolyte might possibly corrode the reactor. Based on the above considerations, a neutral environment was adopted in our NORR experiments. The corresponding data and discussion have been added into the revised manuscript as follows:

“We chose 0.1 M PBS buffer solution to maintain the pH of the electrolyte during NORR.” (Page 13 in the revised manuscript).

“Moreover, we conducted the NORR experiment over Co/Co₃O₄/CoB in 0.1 M KOH aqueous electrolyte. The NORR performance was poorer as compared with the experiment conducted in 0.1 M PBS (Supplementary Fig. 18).” (Page 7 in the revised manuscript).

Fig. R25. The electrochemical NORR performance over Co/Co₃O₄/CoB recorded in 0.1 M PBS and 0.1 M KOH electrolyte.

3. Please provide experimental details for catalyst activation.

Response: Thanks for your valuable suggestion. The details for catalyst activation have been added to the part of “Electrochemical measurements” in Methods, as follows:

“Linear sweep voltammetry was performed at a scanning rate of 5 mV/s prior to 50 cycles of cyclic voltammetry at a scan rate of 50 mV/s to obtain a stable curve. All data presented were within 90% iR-correction. The chronoamperometry was operated to evaluate the stability at different current densities.” (Page 14 in the revised manuscript).

4. In XPS, orbitals should be in italics. The interpretation of XAS and XPS is poor.

Response: We appreciate the reviewer for the helpful comments. Accordingly, we have corrected the writing errors in XPS, as shown in Fig. R26. Furthermore, we have carefully analyzed the data of XAS and XPS and tried our best to give clear interpretations about XAS and XPS results. The corresponding revisions are shown below:

“In Fig. 2b, the Co *2p*_{3/2} XPS spectrum of Co/Co₃O₄ exhibits three peaks at binding energies of 779.7, 781.0 and 783.1 eV, which can be attributed to Co⁰, Co³⁺ and Co²⁺, respectively⁴³⁻⁴⁵. For Co/Co₃O₄/CoB, the Co⁰ peak slightly shifts to a higher binding energy (780.1 eV), while the Co³⁺ (780.7 eV) and Co²⁺ (782.2 eV) peaks shift toward lower binding energies as compared to those for Co/Co₃O₄. Because Co⁰ and Co³⁺/Co²⁺ mainly exist in Co and Co₃O₄, respectively, it can be inferred that the introduction of CoB leads to more electron loss in Co and more electron gain in Co₃O₄. In order to clarify the electron transfer among the interfaces of Co/Co₃O₄/CoB and Co/Co₃O₄, DFT calculations were conducted. Firstly, the Bader charge was analyzed over Co/Co₃O₄ and Co/Co₃O₄/CoB model. For Co/Co₃O₄ model, 3.64 |e| is transferred from Co to Co₃O₄ (Supplementary Fig. 6, Supplementary Table 1). In Co/Co₃O₄/CoB, Co₃O₄ gets 2.23 |e| and 2.36 |e| from Co and CoB,

respectively (Supplementary Fig. 7, Supplementary Table 1). The charge density difference analysis of Co/Co₃O₄/CoB model indicates a significant charge redistribution at the interface of CoB and Co/Co₃O₄ (Fig. 2c), with electron transfer from Co to CoB and from CoB to Co₃O₄. Therefore, among the interfaces of Co/Co₃O₄/CoB, the electrons are transferred from Co to Co₃O₄, Co to CoB, and CoB to Co₃O₄. Furthermore, the average numbers of electron transfer per Co atom in Co/Co₃O₄/CoB and Co/Co₃O₄ were analyzed. As compared in Supplementary Table 1, Co loses more electrons (-0.22 e vs. -0.13 e) and Co₃O₄ gains more electrons (+0.51 e vs. +0.08 e) in Co/Co₃O₄/CoB as compared to Co/Co₃O₄, evidencing that CoB introduction can modulate the electronic structure of Co and Co₃O₄ in Co/Co₃O₄/CoB” (Page 5 in the revised manuscript).

“In the Co K-edge XANES spectra (Fig. 2d), the position of the absorption edge for Co/Co₃O₄/CoB and Co/Co₃O₄ are located between that for Co foil and Co₃O₄ references, indicating that the average valence state of Co atom in Co/Co₃O₄/CoB and Co/Co₃O₄ are within 0 and +2/+3, owing to co-existence of both metallic Co and Co₃O₄. To understand the local structure around Co atom in these samples, the EXAFS spectra are transformed into R-space. The R-space EXAFS spectra (Fig. 2e) display dominant peaks of Co-B (1.60 Å)⁴⁶, Co-Co (2.17Å)⁴⁷, and Co-O (1.44 Å)⁴⁸ coordination bonds in CoB, Co foil and Co₃O₄, respectively. The FT-EXAFS spectra of Co/Co₃O₄/CoB and Co/Co₃O₄ can be well-fitted using backscattering paths of Co-Co, Co-B and Co-O (Fig. 2f, Supplementary Table 2 and 3). Notably, the Co-O coordination number (2.2) in Co/Co₃O₄/CoB is significantly lower than that (5.7) in Co/Co₃O₄. The low-coordinated Co-O structure in Co/Co₃O₄/CoB is beneficial for NO adsorption.⁴⁹ The wavelet transform (WT) contour plot of Co/Co₃O₄/CoB (Fig. 2g) displays three intensity maximums at around 7.2, 5.9, and 5.4 Å⁻¹, corresponding to the Co-Co, Co-B, and Co-O coordination, respectively^{48,50}. For comparison, Co/Co₃O₄ only shows Co-Co and Co-O coordinations (Supplementary Fig. 8).” (Page 5-6 in the revised manuscript).

Fig. R26. Co $2p_{3/2}$ XPS spectra of Co/Co₃O₄ and Co/Co₃O₄/CoB.

5. Zn-NO battery result, y axis should be V vs. Zn/Zn²⁺. Also, there is possibility for formation of ZnO or Zn(OH)₂ during electrochemical battery discharge cycle. Authors, not studied in depth of battery analyses.

Response: We appreciate the reviewer for the constructive comments. We have corrected the Y-axis to V vs. Zn/Zn²⁺, as presented in Fig. R27 (the revised Supplementary Figs. 24-29). In order to confirm

the formation of ZnO or Zn(OH)₂ during electrochemical battery discharge cycle, we conducted XRD measurements of the used Zn plate. As shown in Fig. R28, the XRD results indicated that ZnO was produced on Zn plate after Zn-NO battery testing.

In this study, we mainly investigated the electrocatalytic performance and reaction mechanism of the developed ternary Co/Co₃O₄/CoB catalyst for NORR. The battery research was only an application expansion. As the Zn-NO battery section is not the main research emphasis of this work, we have moved this section to the Supplementary Information (the revised Supplementary Figs. 24-29), and the corresponding discussion has been revised as follows:

“To demonstrate the practical application potential of the Co/Co₃O₄/CoB catalyst, a Zn-NO battery with Co/Co₃O₄/CoB cathode and Zn plate anode was assembled (Supplementary Fig. 24), which could deliver an open circuit voltage (OCV) of 2.04 V (Supplementary Fig. 25) and a maximum power density of 10.06 mW cm⁻², outperforming all reported results in the literature (Supplementary Figs. 26 and 27, Supplementary Table 5). The output discharge current density increases continuously from 0.5 to 8 mA cm⁻² and each step exhibits a stable discharging plateau, indicating that our cell has an excellent discharge capability (Supplementary Fig. 28). Moreover, the NH₃ yield rate exhibits a maximum of 1627.67 μg h⁻¹ mg_{cat}⁻¹ at 8 mA cm⁻² (Supplementary Figure 29). Significantly, the Zn-NO battery can remove NO pollutant, produce NH₃ and generate electricity at the same time.” (Page 7-8 in the revised manuscript).

Fig. R27. **a**, Polarization curve and power density plot of the Zn-NO battery with Co/Co₃O₄/CoB cathode. **b**, OCV of the Zn-NO battery with Co/Co₃O₄/CoB cathode. The inset shows a digital photograph of the Zn-NO battery. **c**, Discharging tests at various current densities.

Fig. R28. The XRD pattern of used Zn plate from Zn-NO battery test.

6. Figure 31, the current density value is negative or positive? Negative for discharge and positive for charging?

Response: We thank the reviewer for raising these questions. As stated by the reviewer, a negative current density corresponds to the discharging process, while a positive value indicates the charging process. In the experiment, we set a negative current to achieve the battery's discharge process. It is worth mentioning that, as stated in the caption of Fig. R29, the figure presents the data of the discharge process under different current densities. We chose not to label the negative current density in the plot itself, as this approach is consistent with the common descriptions in the literature on zinc-air and zinc-NO batteries (Nat. Commun., 2023, 14, 8036; Nat. Commun., 2013, 4, 1805; J. Am. Chem. Soc., 2023, 145, 6899; Angew. Chem. Int. Ed., 2021, 60, 25263; Angew. Chem. Inter. Ed., 2023, 62, e202218717; Energy Environ Sci., 2021, 14, 3938).

Fig. R29. Discharging tests at various current densities.

7. In situ ATR, the surface reconstruction materials must be provided.

Response: Thanks for your valuable suggestion. The surface reconstruction information of the catalyst could not be obtained below the wavenumber of 1000 cm⁻¹ due to the strong noise interference caused by infrared instrument in the *in-situ* ATR-FTIR. In order to confirm if surface reconstruction occurred on the catalyst during NORR, we supplemented the *in-situ* Raman spectra of Co/Co₃O₄/CoB recorded during NORR at different cathodic potentials. As shown in Fig. R30, Co/Co₃O₄/CoB could maintain the Co-O (416 cm⁻¹), B-O (989 cm⁻¹) and H-O-H (1637 cm⁻¹) bending vibration peak strength and position at all potentials, implying the unchanged active phases during NORR (Nat. Commun., 2025, 16, 736; J. Am. Chem. Soc., 2024, 146, 12538; J. Non-Cryst. Solids, 1993, 159, 1). Additionally, we also supplemented the XRD pattern, HRTEM image and XPS spectra of Co/Co₃O₄/CoB after NORR durability test. As shown in Fig. R31 (the revised Supplementary Figure 21), no changes are observed on the two diffraction peaks between fresh and used Co/Co₃O₄/CoB catalyst, indicating no phase change during the NORR. As displayed in the HRTEM image in Fig. R32 (the revised Supplementary Figure 22), the used catalyst still shows the ternary heterostructure. As shown in Fig. R33 (the revised Supplementary Figure 23), the used Co/Co₃O₄/CoB sample shows the similar characteristic peaks of Co⁰, Co²⁺, and Co³⁺ as compared to the fresh one. The above results evidence the good structural stability of Co/Co₃O₄/CoB during NORR, and the corresponding discussion has been added into the revised manuscript as:

“Co/Co₃O₄/CoB shows a good structural stability during the NORR based on the XRD, HRTEM and XPS characterizations (Supplementary Figs. 21-23).” (Page 7 in the revised manuscript).

Fig. R30. The *in-situ* Raman spectra recorded over Co/Co₃O₄/CoB at different cathodic potentials.

Fig. R31. XRD patterns of Co/Co₃O₄/CoB before and after NORR durability test.

Fig. R32. The HRTEM image of Co/Co₃O₄/CoB after NORR durability test.

Fig. R33. Co $2p_{3/2}$ XPS spectra of Co/Co₃O₄/CoB before and after NORR durability test.

8. In the introduction, the importance of NO₂-RR should be emphasized, and recent studies on NORR for ammonia synthesis should be discussed.

Response: Thanks for your valuable suggestion. Our work mainly focuses on NORR and we guess that what you might mean is to include discussion on the significance of NORR. Accordingly, we have revised the introduction as follows:

“One of the alternatives to the Haber-Bosch method for NH₃ synthesis is the electrocatalytic reduction of nitrogen-containing species, including nitrogen gas (N₂), nitric oxide (NO), nitrite (NO₂⁻), and nitrate (NO₃⁻)⁸⁻¹¹. The nitrogen reduction reaction (NRR) suffers from low NH₃ Faradaic efficiency (FE_{NH₃}) and NH₃ yield rate because of the high dissociation energy (941 kJ mol⁻¹) of N≡N and the competitive hydrogen evolution reaction (HER) (the reduction potential of N₂ (0.093 V vs. RHE) and H₂O (0 V vs. RHE) are close to each other)¹²⁻¹⁴. The electroreduction of NO₂⁻ and NO₃⁻ to NH₃ (NO₂⁻RR and NO₃⁻RR) have more complicated reaction pathways, and generate more types of N-containing by-products^{15, 16}. Compared with NRR, the electrochemical NO reduction reaction (NORR) is kinetically and thermodynamically more favorable because NO possesses a lower dissociation energy (204 kJ mol⁻¹) and a more positive reduction potential (0.71 V vs. RHE)¹⁷⁻¹⁹. The approach to couple NO oxidation with NO₃⁻ reduction is also considered as an alternative to NORR. But NORR to produce NH₃ generally requires lower energy consumption^{20, 21}. In recent years, NORR is becoming increasingly attractive because NORR not only can remove environmental pollutant NO from industrial exhaust gas, but also produces value-added chemicals.” (Page 2 in the revised manuscript).

“To improve the NORR performance, researchers have made endeavors to enhance the adsorption of NO by transition metal oxide modification or vacancy engineering.^{25, 26} The proton supply could be enhanced by using metal-based electrocatalysts that would facilitate H₂O dissociation^{24, 27}. Constructing single atoms in amorphous metal oxides with oxygen vacancies to form metal-O moieties could accelerate hydrogenation of NO to produce NH₃.²⁸⁻³⁰ However, till now, it is still intractable to acquire both high FE_{NH₃} and NH₃ yield rate simultaneously because of the complex balance of the NO adsorption and H₂O dissociation.” (Page 2 in the revised manuscript).

9. It is suggested to include a comparison with other state-of-the-art catalysts in the discussion section.

Response: Thanks for your valuable suggestion. We have conducted a performance comparison with the state-of-the-art NORR electrocatalysts. As shown in Fig. R34 (Figure 3f in the revised manuscript) and Table R5 (Supplementary Table 4), the ternary Co/Co₃O₄/CoB is able to achieve a high FE_{NH₃} of 98.8% with an NH₃ yield rate of 462.18 μmol h⁻¹ cm⁻² as well as long-time stability, superior to most of the reported NORR electrocatalysts. The corresponding discussion has been updated in the revised manuscript, as follows:

“The ternary Co/Co₃O₄/CoB achieves a high FE_{NH₃} of 98.8% and NH₃ yield rate of 462.18 μmol h⁻¹ cm⁻² as well as long-time stability at a low cathodic potential of -0.5 V vs. RHE, superior to most of the reported NORR electrocatalysts (Fig. 3f and Supplementary Table 4).” (Page 7 in the revised manuscript).

Fig. R34. Comparison of electrocatalytic NORR performance of Co/Co₃O₄/CoB with other reported electrocatalysts in literature.

Table R5. Comparison of NH₃ yield rate and FE_{NH₃} of Co/Co₃O₄/CoB with the reported NORR electrocatalysts.

Catalyst	Electrolyte	NH ₃ yield rate (μmol·h ⁻¹ ·cm ⁻²)	FE _{NH₃} (%)	Stability time (h)	V vs. RHE	References
Co/Co ₃ O ₄ /CoB	0.1 M PBS	462.18	98.80	~100	-0.5	This work
hcp-Co	0.1 M Na ₂ SO ₄	439.50	72.58	6	-0.6	R1
Nb-SA/BNC	0.1 M Na ₂ SO ₄	295.2	48.14	56	-0.9	R2
Cu (111)	0.1 M Na ₂ SO ₄	187.5	93.19	5	-0.59	R3
a-B _{2.6} C@TiO ₂ /Ti	0.1 M Na ₂ SO ₄	216.39	87.6	12	-0.9	R4
Fe ₁ /MoS _{2-x}	0.1 M Na ₂ SO ₄	288.2	82.5	15	-0.6	R5
Sb ₁ /a-MoO ₃	0.1 M Na ₂ SO ₄	273.5	91.7	30	-0.6	R6

Cu ₁ /MoS ₂	0.1 M Na ₂ SO ₄	337.5	90.6	20	-0.6	R7
NiO/TM	0.1 M Na ₂ SO ₄	125.29	90	12	-0.6	R8
CoB/Co@C	0.5 M Na ₂ SO ₄	315.4	~ 85	10	-0.6	R9
Sb ₂ S ₃	0.1 M Na ₂ SO ₄	168.6	93.7	20	-0.7	R10

References

- R1. Wang, D. et al. Hexagonal cobalt nanosheets for high-performance electrocatalytic NO reduction to NH₃. *J. Am. Chem. Soc.* **145**, 6899-6904 (2023).
- R2. Peng, X. et al. Ambient electrosynthesis of ammonia with efficient denitration. *Nano Energy*. **78**, 105321 (2020).
- R3. Xiao, L. et al. Identification of Cu(111) as superior active sites for electrocatalytic NO reduction to NH₃ with high single-pass conversion efficiency. *Angew. Chem. Int. Ed.* **63**, e202319135 (2024).
- R4. Liang, J. et al. Amorphous boron carbide on titanium dioxide nanobelt arrays for high-efficiency electrocatalytic NO reduction to NH₃. *Angew. Chem. Int. Ed.* **61**, e202202087 (2022).
- R5. Chen, K. et al. Atomically Fe-doped MoS_{2-x} with Fe-Mo dual sites for efficient electrocatalytic NO reduction to NH₃. *Appl. Catal. B: Environ.* **324**, 122241 (2023).
- R6. Chen, K. et al. p-Block antimony single-atom catalysts for nitric oxide electroreduction to ammonia. *ACS Energy Lett.* **8**, 1281-1288 (2023).
- R7. Chen, K., Zhang, G., Li, X., Zhao, X. & Chu, K. Electrochemical NO reduction to NH₃ on Cu single atom catalyst. *Nano Res.* **16**, 5857-5863 (2023).
- R8. Liu, P. et al. High-performance NH₃ production via NO electroreduction over a NiO nanosheet array. *Chem. Commun.* **57**, 13562-13565 (2021).
- R9. Wu, B. et al. Boron-modulated electronic-configuration tuning of cobalt for enhanced nitric oxide fixation to ammonia. *Nano Lett.* **23**, 7120-7128 (2023).
- R10. Chen, K., Zhang, Y., Du, W., Guo, Y. & Chu, K. Atomically isolated and unsaturated Sb sites created on Sb₂S₃ for highly selective NO electroreduction to NH₃. *Inorg. Chem. Front.* **10**, 2708-2715 (2023).

10. Besides NH₃, it is essential to thoroughly examine the formation of other products.

Response: Thanks for your valuable suggestion. The gas phase and liquid phase products have been carefully quantified:

(1) Liquid phase by-products

NH₂OH is the most likely liquid-phase by-product in NORR. To quantify NH₂OH, the following procedures were executed: 1.0 mL 0.1~0.5 mM standard NH₂OH solution was added with 1.0 mL PBS buffer (pH 7.4) and 1.0 mL 1% 8-hydroxyquinoline. Under vigorous shaking, 1.0 mL 0.1 M Na₂CO₃ was added and the mixture was heated at 100 °C for 1 min. In the presence of NH₂OH, the solution would turn from light yellow to blue-green, showing an absorption peak at ~705 nm in the UV-vis absorption spectra. Plotting the peak absorbance with different NH₂OH concentration yielded a calibration curve, as seen in Figs. R35 and R36 (Supplementary Figure 14 and 15). After 1 h of NORR

at -0.5 V vs. RHE, as shown in Fig. R37 (Supplementary Figure 16), no peaks were detected at 705 nm, ruling out the formation of NH_2OH over $\text{Co}/\text{Co}_3\text{O}_4/\text{CoB}$ and $\text{Co}/\text{Co}_3\text{O}_4$ during NORR.

(2) Gas phase by-products

After NORR, we collected the exhaust gas using a gas bag and checked it using a gas chromatography. The gas phase by-products including N_2O and N_2 were not detected. From the semi-quantitative analysis based on online differential electrochemical mass spectrometry (DEMS, Fig. R38, Supplementary Fig. 17), it was found that the NORR over $\text{Co}/\text{Co}_3\text{O}_4/\text{CoB}$ did not produce N_2 and N_2O , but produced a very small amount of H_2 (an order of magnitude lower than the main product NH_3). The corresponding results and discussion have been added into the revised manuscript and Supplementary Information as:

“The possible by-products produced over $\text{Co}/\text{Co}_3\text{O}_4/\text{CoB}$ during NORR were checked by differential electrochemical mass spectroscopy (DEMS), UV-vis absorption spectroscopy, and gas chromatography (Supplementary Figs. 14-17). The hydroxylamine (NH_2OH), N_2 and N_2O by-products were not detected.” (Page 7 in the revised manuscript).

Fig. R35. UV-vis absorption spectra of NH_2OH solution with concentrations in the range of 0-0.5 mM in 0.1 M PBS.

Fig. R36. The linear calibration curve for quantifying NH_2OH .

Fig. R37. UV-vis absorption spectra of the catholyte collected after 1 h of NORR at -0.5 V vs. RHE.

Fig. R38. The DEMS signals.

11. To further support their findings, the authors should include *in situ* Raman spectra at different applied potentials.

Response: We thank the reviewer for the valuable suggestion. Based on which, we have supplemented *in-situ* Raman spectra recorded during NORR at different applied cathodic potentials. As displayed in Figs. R39 and R40 (Supplementary Figs. 37 and 38), the *in-situ* Raman spectra indicated the influence of interfacial H₂O on electrochemical NORR over Co/Co₃O₄/CoB and Co/Co₃O₄. In neutral media, the interfacial H₂O over catalyst provides the proton source for the hydrogenation reaction. We performed Gaussian fitting to the broad Raman band of the O-H stretching, which could be resolved into three distinct components, corresponding to three types of O-H stretching belonging to 4-HB·H₂O, 2-HB·H₂O, and M·H₂O (M = Na⁺/K⁺), respectively (Nature, 2021, 81, 600). The relative proportion of M·H₂O displayed a little increasing trend along with increasing the applied cathodic potential, suggesting that M·H₂O was closer to the catalyst's surface and could be more affected by the electric field. The M·H₂O proportion over Co/Co₃O₄/CoB is always higher than that over Co/Co₃O₄ at all studied cathodic potentials, suggesting better water dissociation ability over Co/Co₃O₄/CoB (Nat. Commun., 2023, 14, 5289; Adv. Energy Mater., 2024, 14, 2400065). We have supplemented the

experimental results and added the corresponding discussion into the revised manuscript and Supplementary Information as follows:

“In order to further investigate the H₂O dissociation ability of Co/Co₃O₄/CoB and Co/Co₃O₄, *in-situ* Raman spectroscopy measurements were conducted at different applied cathodic potentials. Three types of O-H stretching modes belonging to 4-HB·H₂O, 2-HB·H₂O, and M·H₂O (M = Na⁺/K⁺), respectively,⁵⁵ were observed. The relative proportion of M·H₂O displayed a little increasing trend along with increasing the applied cathodic potential, suggesting that M·H₂O was closer to the catalyst’s surface and could be more affected by the electric field. The M·H₂O proportion over Co/Co₃O₄/CoB is always higher than that over Co/Co₃O₄ at all studied cathodic potentials, suggesting better water dissociation ability over Co/Co₃O₄/CoB (Supplementary Figs. 37 and 38)^{56, 57}.” (Page 9 in the revised manuscript).

Fig. R39. The *in-situ* Raman spectra recorded over Co/Co₃O₄/CoB at different applied cathodic potentials (M refers to Na⁺ or K⁺).

Fig. R40. The *in-situ* Raman spectra recorded over Co/Co₃O₄ at different applied cathodic potentials (M refers to Na⁺ or K⁺).

Reviewer #4:

In this work, Fan et al. prepared a ternary heterostructured Co/Co₃O₄/CoB catalyst for electrochemical NO reduction. The catalyst exhibits superior performance over the reported ones due to the synergistic roles of these three phases with various activity towards the H₂O splitting, NO adsorption, as well as H* transfer. The DFT calculations have been conducted to support the experimental results. Overall, this is an interesting work regarding the NO removal and catalyst design. However, to the best of my knowledge, the simulations seem to be not rigorous to support the results and hypothesis. I suggest major revision after addressing the follows comments.

Response: We thank the reviewer for the time and efforts spent in assessing our manuscript. Based on the reviewer's valuable comments and suggestions, we have conducted additional experiments and DFT simulations to clarify the concerns raised by the reviewer. Below please find our point-to-point response letter.

1. From the Fig. 1d, the catalyst shows three boundary between Co (111), CoB (210), and Co₃O₄ (311) surfaces. However, they applied a CoB supported (Co₃O₄-Co) cluster model to represent the catalyst. More specifically,

i) Why is this type of model applied? As the cluster did not contain any crystal information about the Co₃O₄ and Co (111) surfaces.

Response: We appreciate the reviewer for raising the question. To construct the appropriate calculation models, first, we determined the morphology of Co/Co₃O₄/CoB heterostructure, which composed of a CoB nanosheet supported with large Co and small Co₃O₄ nanoparticles based on electron energy loss spectroscopy (EELS) mapping (Fig. R41 and Fig. 1e) and HRTEM (Fig. R42 and Supplementary Figure 2) measurements. Such a ternary Co/Co₃O₄/CoB heterostructure contained three interfaces of Co-Co₃O₄, Co-CoB, and Co₃O₄-CoB. Considering the morphology and three interfaces of the Co/Co₃O₄/CoB heterostructure, we opted for a simplified model, namely the neighbouring Co and Co₃O₄ clusters supported on CoB, to elucidate the roles of ternary heterostructure in electrochemical NO reduction to NH₃. The metal/metal oxide nanoparticles were modeled by clusters, which is a common method in DFT calculations used in the literature (Nat. Commun., 2024, 15, 8444; Nat. Catal., 2021, 4, 1032; Nat. Commun., 2019, 10, 1166). The choice of crystal facets of Co (111), Co₃O₄ (311), and CoB (021) was primarily based on the strongest diffraction peak of Co, Co₃O₄ and CoB observed in the XRD pattern (Fig. R43 and Fig. 2a). These crystal facets are also thermodynamically stable. It is noted that the Co (111) and Co₃O₄ (311) clusters are taken from (111) facet of Co single crystal and (311) facet of Co₃O₄ single crystal, respectively. Accordingly, CoB (021) with supported neighbouring Co (111) and Co₃O₄ (311) clusters was adopted to represent the Co/Co₃O₄/CoB heterostructure (Fig. R44). The corresponding discussion has been added into the revised manuscript as:

“Herein, by balancing the computation power and calculation accuracy, the Co/Co₃O₄/CoB was modeled by Co (111) and Co₃O₄(311) clusters supported on CoB (021), while Co/Co₃O₄ was modeled by Co (111) clusters supported on Co₃O₄ (311) (for details see DFT calculations in Method section).” (Page 5 in the revised manuscript)

“Considering the limitations of computational power, we opted for a simplified mode, namely CoB supported neighboring Co and Co₃O₄ clusters, combining with the morphology and three interfaces of Co/Co₃O₄/CoB heterostructure. The choice of using crystal facets of Co (111), Co₃O₄ (311), and CoB (021) to model Co/Co₃O₄/CoB heterostructure is primarily based on the XRD characterizations.

These crystal facets exhibit the strongest diffraction in the XRD patterns and are also the thermodynamically stable facets. It is noted that the Co (111) and Co_3O_4 (311) clusters are taken from (111) facet of Co single crystal and (311) facet of Co_3O_4 single crystal, respectively.” (Page 17 in the revised manuscript).

Fig. R41. HAADF-STEM image and the corresponding EELS mappings of Co, O, and B elements in $\text{Co}/\text{Co}_3\text{O}_4/\text{CoB}$.

Fig. R42. The HRTEM image of $\text{Co}/\text{Co}_3\text{O}_4/\text{CoB}$.

Fig. R43. XRD patterns of $\text{Co}/\text{Co}_3\text{O}_4$ and $\text{Co}/\text{Co}_3\text{O}_4/\text{CoB}$.

Fig. R44. The optimized model of Co/Co₃O₄/CoB.

ii) How to ensure the stability of this model? Did the authors conduct AIMD to validate it? Based on the illustration in Supplementary Figure 6, the configuration seems to be not well relaxed.

Response: Thanks for your valuable suggestion. In the “Density functional theory (DFT) calculations” part in our original manuscript, we have stated that “All the structures were relaxed until the residual forces on the atoms had declined to less than 0.05 eV/Å”. The maximum atomic force in Co/Co₃O₄/CoB is 0.047, which meets the convergence requirement. In addition, we have also evaluated the structural stability by using AIMD (Figs. R45-R47). In the AIMD simulation up to 10 ps, the heterostructure of Co/Co₃O₄/CoB was not destroyed, and the basic crystal structure was stable, proving the rationality of the structure. The corresponding results and discussion have been added into the revised manuscript and Supplementary Information as:

“The maximum atomic force in Co/Co₃O₄/CoB is 0.047, which meets the convergence requirement. In addition, we also evaluated the structural stability by using ab initio molecular dynamics (AIMD). The Brillouin zone was sampled using $1 \times 1 \times 1$ k-point grid. Self-consistent calculations were conducted with an energy convergence threshold of 10^{-5} eV. The AIMD was performed within the canonical (NVT) ensemble by Nosé-Hoover thermostats with a time step of 1.0 fs at a finite temperature of 300 K⁷⁰. In the AIMD simulation up to 10 ps, the heterostructure of Co/Co₃O₄/CoB was not destroyed, and the basic crystal structure was stable, proving the rationality of the structure (Supplementary Fig. 55-57).” (Page 17 in the revised manuscript).

Fig. R45. Variations of temperature during the AIMD simulation for assessing the stability of the Co/Co₃O₄/CoB model.

Fig. R46. Variations of potential energy during the AIMD simulation for assessing the stability of the Co/Co₃O₄/CoB model.

The potential energy of the structure dropped sharply within the first 1 ps, releasing heat and undergoing relaxation. After stabilizing for 3 ps from 1 to 4 ps, the structure experienced a slight relaxation again. Over the extended period from 5 to 10 ps, the potential energy of the structure converged, and the structure reached a pre-equilibrium state.

Fig. R47. The structure mode of Co/Co₃O₄/CoB in 0 ps to 10 ps.

It could be observed that during the 10 ps AIMD simulation, the Co/Co₃O₄/CoB interface was not disrupted.

iii) What are the oxidation states of the Co₃O₄ phase in this model? 2+ or 3+? How were the magnetizations determined in this model? Why DFT+U is not applied to the Co₃O₄?

Response: We appreciate the reviewer for raising these questions. Based on XPS measurements (Fig. R48 and Fig. 2b in the revised manuscript), the Co₃O₄ showed two types of Co valence: Co²⁺ and Co³⁺. The influence of spin of Co has been considered in the DFT calculation model. Because there is a lack of uniform understanding on the setting of magnetic moment, the default setting of magnetic moment (MAGMOM= number of atoms of Co*1.0) was chosen for all calculations. The calculated results show that the total magnetic moment of the system is 44, and the number of Co atoms in the system is 78. For the magnetic moment less than 1.0, it is reasonable to use the default magnetic moment. We have added the relevant discussion into the revised manuscript:

“Spin polarization was included in the calculations and the default setting of magnetic moment (MAGMOM = number of atoms of Co*1.0) was chosen for all calculations.” (Page 17 in the revised

manuscript).

It is well known that the simulation results of bandgap in DFT calculation differ greatly from the experimental results. Therefore, a +U correction is generally considered necessary. However, our calculation process mainly focused on the energy of the system and did not involve the analysis of the electronic structure of the system with strongly correlated energy band. Therefore, DFT+U was not considered.

Fig. R48. Co $2p_{3/2}$ XPS spectra of Co/Co₃O₄ and Co/Co₃O₄/CoB.

2. Since three surfaces are observed in the experiments, obviously, the synergistic interplay of these three surfaces balances the RDS and promotes the NORR. The authors should conduct the DFT calculations on each of the surface to explore the rate limiting steps and investigate how synergistic mechanism works on the boundary. Assuming that the diffusion steps are not rate limiting steps, the three phases might play roles in different stages of NORR reaction network, similar to the most recent work of Wang et al (Small 2024, 20, 2311439).

Response: We thank the reviewer for the nice suggestion. According to which, we have supplemented the DFT calculations for NORR taking place over Co, CoB, and Co₃O₄. As displayed in Fig. R49, all three samples show two energy barriers, namely NO* to NOH* and NHOH* to NH*. The ΔG of NO* to NOH* is 2.81, 1.6, and 1.25 while that of NHOH* to NH* is 0.31, 0.81, 0.30 over Co, CoB, and Co₃O₄, respectively. Among the three phases, Co₃O₄ is most thermodynamically favorable for both two steps, which can serve as the dominant sites for NORR. Therefore, it can be excluded that the three phases play roles in different stages of NORR. Furthermore, the DFT calculations were also performed on Co₃O₄ in the Co/Co₃O₄/CoB and Co/Co₃O₄ model. The hydrogenation of NO* to NOH* is the potential-determining step (PDS) of NORR on Co/Co₃O₄/CoB and Co/Co₃O₄, and the former one has a lower reaction energy barrier owing to the enhanced charge and proton transfer over the ternary Co/Co₃O₄/CoB heterostructure.

“Besides, the reaction energy of NORR was also calculated over Co₃O₄, Co, and CoB (Supplementary Figs. 51-54 and Table 7). The reaction energy of the potential-determining step (*NO to *HNO) is 1.25, 2.81 and 1.6 eV over Co₃O₄, Co, and CoB, respectively, much higher than that over Co/Co₃O₄/CoB. This result evidences that the first step of NO hydrogenation can be notably boosted over the ternary Co/Co₃O₄/CoB heterostructure.” (Page 12 in the revised manuscript).

Fig. R49. The calculated NORR Gibbs free energy diagrams over Co_3O_4 , Co, CoB, Co/Co $_3\text{O}_4$ and Co/Co $_3\text{O}_4$ /CoB at 0 V vs. RHE.

Reviewer #5:

Response: Thanks for your valuable comments. We have replied all the comments and suggestions raised by all reviewers point-by-point.